DISCOVERY REPORT

# The absence of core piRNA biogenesis factors does not impact efficient transposon silencing in *Drosophila*

**Shashank Chary[1], Rippei Hayashi** [1,2]*

**1** John Curtin School of Medical Research, The Australian National University, Acton, Australian Capital Territory, Australia, **2** The Shine-Dalgarno Centre for RNA Innovation, The Australian National University, Acton, Australian Capital Territory, Australia

* rippei.hayashi@anu.edu.au

**Data Availability Statement:** Sequencing data and processed files have been deposited to Gene

## Abstract

Organisms require mechanisms to distinguish self and non-self-RNA. This distinction is crucial to initiate the biogenesis of Piwi-interacting RNAs (piRNAs). In *Drosophila* ovaries, PIWI-guided slicing and the recognition of piRNA precursor transcripts by the DEAD-box RNA helicase Yb are the 2 known mechanisms to licence an RNA for piRNA biogenesis in the germline and the soma, respectively. Both the PIWI proteins and Yb are highly conserved across most *Drosophila* species and are thought to be essential to the piRNA pathway and for silencing transposons. However, we find that species closely related to *Drosophila melanogaster* have lost the *yb* gene, as well as the PIWI gene *Ago3*. We show that the precursor RNA is still selected in the absence of Yb to abundantly generate transposon antisense piRNAs in the soma. We further demonstrate that *Drosophila eugracilis*, which lacks *Ago3*, is completely devoid of ping-pong piRNAs and exclusively produces phased piRNAs in the absence of slicing. Thus, core piRNA pathway genes can be lost in evolution while still maintaining efficient transposon silencing.

## Introduction

The ever-changing landscape of transposable elements (TEs) in the genome requires the host defence pathway to be highly plastic and adaptive to maintain silencing. One such mechanism is the Piwi-interacting RNA (piRNA) silencing pathway. piRNA is a class of small RNA of 25 to 32 nucleotides in length and is loaded onto the PIWI-clade Argonaute proteins to form the piRNA-induced silencing complex (pi-RISC) [1,2]. piRNAs are highly diverse in sequence yet are highly specific to TE sequences. Mechanisms underlying the distinction between TEs and host genes remain to be fully understood.

The piRNA pathway is predominantly active in the germline lineage, protecting genome integrity and gonadal development from harmful TE insertions [3]. Some TEs can be uniquely expressed in somatic cells and can carry genes that allow them to infect other cells [4,5]. One notable example are *gypsy* retrotransposons in *Drosophila* that carry retroviral envelope (*env*) genes [6], which are expressed in the ovarian somatic cells (OSCs) and can infect the germline

Expression Omnibus (GSE213383). Codes for the computational analyses are available at https://github.com/RippeiHayashi/noYb_Dspp.git.

**Funding:** This work was supported by Australian Research Council (https://www.arc.gov.au/, DP210102385) to RH, Australian National University start-up (https://www.anu.edu.au/) to RH, and Australian National University Vice-Chancellor's travel fellowship (https://www.anu.edu.au/study/scholarships/find-a-scholarship/vice-chancellors-hdr-travel-grants) to SC. The funders had no role in study design, data collection and analysis, decision to publish, or preparation of the manuscript.

**Competing interests:** The authors have declared that no competing interests exist.

**Abbreviations:** ChIP, chromatin immunoprecipitation; FISH, fluorescent in situ hybridisation; OSC, ovarian somatic cell; piRNA, Piwi-interacting RNA; pi-RISC, piRNA-induced silencing complex; RDC, Rhino-Cutoff-Deadlock; TE, transposable element; tRNA, transfer RNA.

cells. As such, *Drosophila* species express piRNAs as well as piRNA pathway proteins in both the somatic and germline cells of the ovaries [7].

piRNAs are abundantly produced from discrete genomic loci called piRNA clusters. In *Drosophila* ovaries, germline clusters are comprised of TE insertions in both orientations and produce piRNAs from both strands by virtue of noncanonical convergent transcription [8,9]. In contrast, the somatic cluster *flamenco* in *D. melanogaster* consists of inverted repeats of *gypsy* retrotransposons and is transcribed from a canonical RNA Polymerase II promoter and predominantly produces *gypsy* antisense piRNAs [10,11].

There are 2 distinct modes of piRNA biogenesis, called "ping-pong" and "phasing." Ping-pong biogenesis starts with the cleavage (termed slicing) of a precursor transcript by a pi-RISC, which generates the 5′ end of a secondary piRNA. The secondary pi-RISC, in turn, cleaves a transcript from the opposite strand to generate another piRNA, thereby amplifying both sense and antisense piRNA pools [10,12]. Phasing biogenesis (also called primary biogenesis) starts with 5′ end of a piRNA precursor transcript to be loaded onto a PIWI protein, which is followed by the head to tail fragmentation into mature piRNAs by the mitochondrial endonuclease Zucchini/MitoPLD [13–16]. Both ping-pong and phasing mechanisms are evolutionarily highly conserved, present in sponge, sea anemone, and hydra species all the way to humans [17–19].

*Drosophila* expresses 3 PIWI proteins in the ovaries, Piwi, Aubergine, and Ago3. The somatic niche only expresses Piwi, while the germline cells express all three [10]. Ping-pong biogenesis predominantly occurs between Aubergine and Ago3, while Piwi and Aubergine can receive phasing piRNAs. Ping-pong biogenesis is, therefore, specific to the germline, while phasing can happen in either cell niche.

For both ping-pong and phasing, a mechanism to select and trigger piRNA biogenesis is necessary to specifically produce TE antisense piRNAs. The ping-pong mechanism itself provides the solution via the production and binding of sense piRNAs, which target and enrich for TE antisense piRNAs via sequence complementarity. Phasing, however, requires a "triggering" event instead [20]. One way to trigger phasing in the *Drosophila* ovaries is via slicing. The same pi-RISC that is produced in the ping-pong cycle initiates phasing by cleaving a transcript and releasing the 5′ end of a piRNA precursor. This slicer activity has been demonstrated to be the main source of triggering phasing in the *Drosophila* germline [21]. This mechanism of triggering appears to be evolutionarily conserved, as mouse pachytene piRNAs also require slicer activity to start phased biogenesis [22,23].

In the *Drosophila* soma, where ping-pong is absent, the recruitment of Zucchini by the TDRD12 homologue Yb mediates phasing instead. Yb forms an organelle called Yb body at the nuclear periphery to recruit piRNA biogenesis factors, such as the RNA helicase Armitage, to the *flamenco*-derived piRNA precursor transcripts [24,25]. While Armitage is sufficient to induce piRNA biogenesis independently [26], Yb is required for the preferential production of piRNAs from the *flamenco* locus. *Drosophila* mutants for *yb* increase promiscuous piRNA production, which up-regulates transposons and leads to female infertility [25,27,28].

Given that these mechanisms ensure the production of TE antisense piRNAs, the proteins associated with these processes are likely highly conserved [29]. Indeed, the 3 PIWI proteins, Yb, and Armitage are also present in ancient *Drosophila* species such as *Drosophila virilis* and *Drosophila mojavensis*.

However, our inspection on FlyBase (https://flybase.org/) failed to identify Yb homologues in the *obscura* group within *Drosophila*, as well as *Drosophila eugracilis*. More strikingly, we also could not find a homologue of Ago3 in *D. eugracilis*. We characterised ovarian piRNA populations in these species, finding that the *obscura* group as well as *D. eugracilis* predominantly produce TE antisense piRNAs in the soma despite the lack of Yb. We further find that

ping-pong biogenesis is entirely absent in *D. eugracilis* and that TE antisense piRNAs are produced by phasing without slicing in the germline. The study therefore reveals novel routes by which TE antisense transcripts are selected for piRNA production.

## Results

### *yb* gene is lost in species of the *D. obscura* group and *D. eugracilis*

In *Drosophila melanogaster*, the depletion of Yb disrupts the formation of the processing body and disperses other biogenesis factors in the cytoplasm [30,31] (Fig 1A). This results in a marked decrease in the level of *flamenco*-derived piRNAs and the overexpression of *gypsy* transposons by more than 100-fold [25,27] (Fig 1B).

Yb is encoded by the *female sterile 1 yb* (*fs(1)yb*) gene (named *yb* gene hereafter) in *D. melanogaster*. Unexpectedly, FlyBase did not annotate the *yb* gene in some of the most commonly studied *Drosophila* species, including *Drosophila pseudoobscura* and *Drosophila persimilis* from the *obscura* group. To extend this observation, we examined the conservation of *yb* in all *Drosophila* species whose genome sequences were available in high quality (see methods). This survey revealed that nearly all *Drosophila* species outside the *obscura* group species carry homologues of *yb* (Fig 1C and S1 Data). Importantly, the N-terminally located Hel-C domain was found in all *yb* homologues, suggesting the functional conservation of *yb* gene across *Drosophila* species. Strikingly, none of the genomes from the 12 *obscura* group species analysed contained *yb* homologues, whereas homologues of *boyb* and/or *soyb* were found, demonstrating the lack of *yb* gene in the *obscura* group species (S1 Data). Of *Drosophila* species outside the *obscura* group, we also failed to find the *yb* homologues in 3 independent genome assemblies of *D. eugracilis*. Crucially, flanking genes of *yb* are present at the syntenic locus of the *D. pseudoobscura* and *D. eugracilis* genomes (Fig 1D). These genes are also found next to the *yb* homologues of more ancient *Drosophila* species *D. willistoni* and *D. virilis* (S1B and S1C Fig), indicating that the *yb* gene has been specifically lost in these species. We also found that the fragments of domains in Yb are still present in the *D. eugracilis* genome (Fig 1E), but no mRNA expression for these fragments was detected by RNA sequencings in the ovaries (S1A Fig), indicating that it has recently become a pseudogene. The losses of *yb* in the *obscura* group and *D. eugracilis* likely occurred independently, as species that are evolutionarily closer to *D. eugracilis* than the *obscura* group species, have intact *yb* genes (Fig 1C).

### *gypsy-env* retrotransposons are present in species that have lost *yb*

The absence of *yb* in the *obscura* group and *D. eugracilis* prompted us to search for full-length copies of *gypsy* retrotransposons that contain *env* genes (*gypsy*-env), which are normally silenced by Yb-dependent piRNAs. Using existing annotations on RepBase, we identified several *gypsy* elements that carry sequences homologous to *env*, in *D. pseudoobscura*, *D. eugracilis*, as well as 2 other *obscura* group species, *Drosophila bifasciata* and *Drosophila azteca* (see methods). We also observed that each identified *gypsy-env* had multiple intact copies in their genomes (S2 and S3 Data files). These observations strongly suggest that active copies of the *gypsy-env* retrotransposons remain in the *obscura* group species and in *D. eugracilis*, despite the absence of *yb*.

### Both germline and somatic PIWI are present in the *obscura* group species and *D. eugracilis*

The lack of *yb* also prompted us to examine the expression of the PIWI proteins. A tBlastn search confirmed that all 3 PIWI genes, *piwi*, *aubergine*, and *Ago3*, are present in the *obscura*

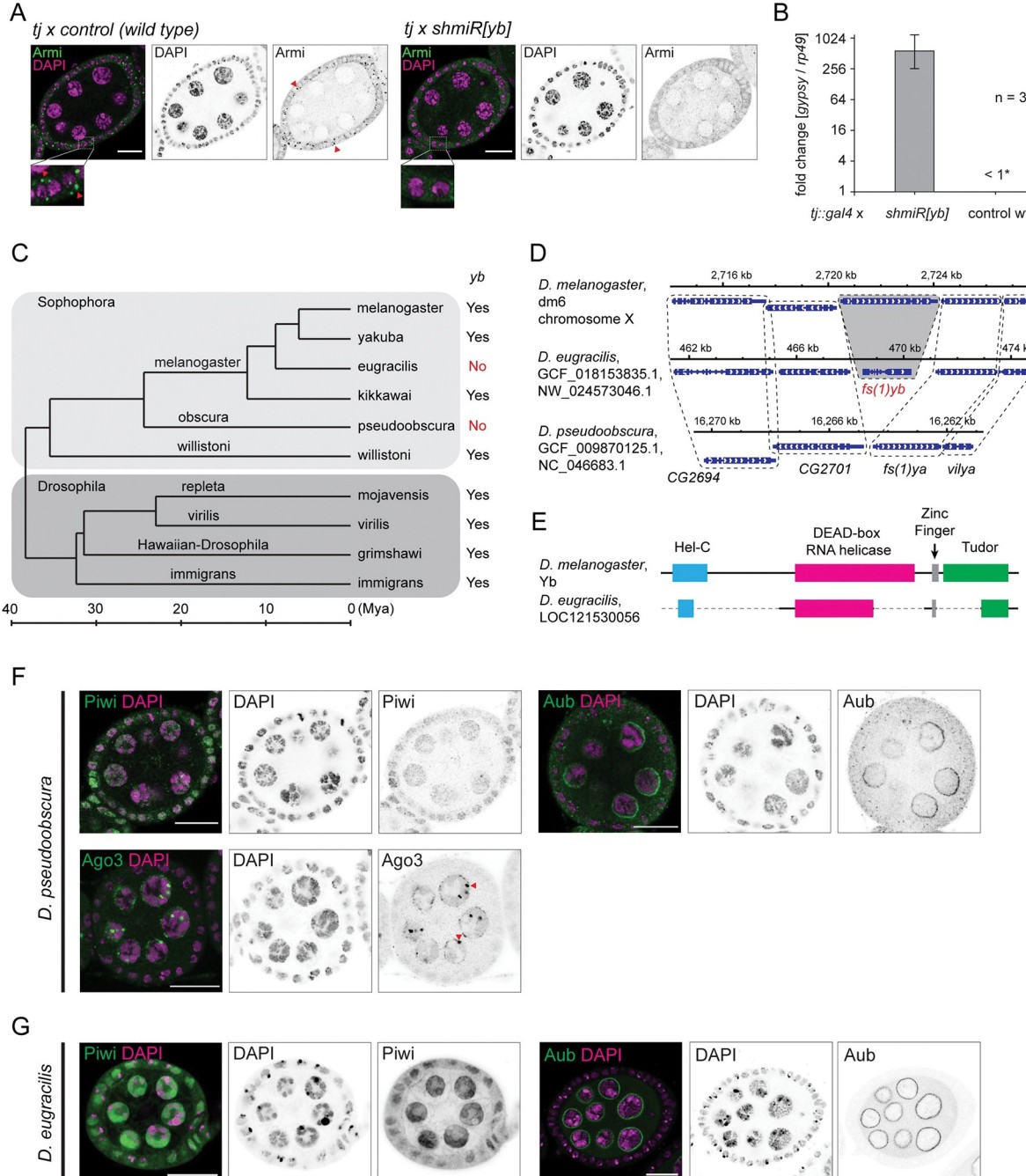

**Fig 1. Conserved localisation of PIWI proteins despite loss of *fs(1)yb* (*yb*) in the *obscura* group and in *D. eugracilis*. (A)** Immunofluorescent staining of Armitage (Armi in green and DAPI in magenta) in the Yb-depleted (tj::gal4 × shmiR[yb]) and the wild-type control (tj::gal4 × control) egg chambers shows that the focal localisation of Armi in the somatic cells (marked by read arrowheads) requires Yb. Enlarged images also highlight the Armi foci in the wild-type somatic cells. Scale bars = 20 μm (**B**) An RT-qPCR shows that the depletion of Yb in the somatic cells results in an overexpression of *gypsy* (see methods). (**C**) The phylogenetic tree of selected *Drosophila* species showing the conservation of *yb* gene. The subgenera *Sophophora* and *Drosophila* are grouped in boxes, and species groups are indicated on branches. Years of divergence and distances between species are based on previous studies [64,65]. (**D**) The conservation of the neighbouring genes in the syntenic locus of *yb* shows the degeneration and loss of *yb* gene in *D. eugracilis* and *D. pseudoobscura*, respectively. (**E**) The degeneration of *yb* gene in *D. eugracilis* is shown alongside with the functional domains. (**F**) Immunofluorescent staining of Piwi, Aubergine, and Ago3 in *D. pseudoobscura* egg chambers (antibodies in green and DAPI in magenta) showing the nuclear localisation of Piwi in both somatic and germline cells, and the perinuclear localisation of Aubergine and Ago3 only in the germline cells. Large perinuclear granules of Ago3 are indicated by arrowheads. (**G**) Immunofluorescent staining of Piwi and Aubergine in *D. eugracilis* egg chambers (antibodies in green and DAPI in magenta) showing the nuclear localisation of Piwi in both somatic and germline cells and the perinuclear localisation of Aubergine. Scale bars = 20 μm. The underlying data can be found in S6 Data file.

group species, with *aubergine* being duplicated or triplicated in some of them (summarised in S4 Data). However, a tBlastn search of Ago3 homologues identified the homologue of Aubergine as the top hit in the 3 independently assembled *D. eugracilis* genomes, suggesting the absence of the *Ago3* gene in this species. The consequences of the apparent loss of *Ago3* will be described in the later part of this work. We then analysed the localizations of Piwi, Aubergine, and Ago3 in *D. pseudoobscura* and *D. eugracilis* ovaries with antibodies raised against Piwi, Aubergine, and Ago3 (Fig 1F and 1G; see methods). Piwi was nuclear both in the somatic and germline cells. Aubergine and Ago3 were only expressed in the germline and localised to the nuclear periphery. Ago3 also formed large granules near the nurse cell nuclei in *D. pseudoobscura* ovaries (Fig 1F) as well as in other *obscura* group species (S2C Fig). Additionally, Aubergine and Piwi localise to the pole cells and other somatic nuclei in the *D. eugracilis* blastoderm-stage embryos (S2D Fig). These observations suggest that both the germline and the somatic piRNA pathways are active in the *obscura* group species and *D. eugracilis*, and their PIWI proteins function similarly to those in *D. melanogaster*.

### *flamenco*-like uni-strand clusters produce abundant piRNAs in the *obscura* group species

The *flamenco* locus in *D. melanogaster* predominantly consists of inverted repeats of *gypsy* retrotransposons. We investigated whether similar loci exist in species that do not have *yb* and, if so, whether they make abundant piRNAs in the soma. We sequenced oxidised small RNA libraries from whole ovaries of the 2 *obscura* group species *D. pseudoobscura* and *D. bifasciata*. Inspections of small RNAs that uniquely mapped to the genome coupled with transposons predicted by RepeatMasker identified *flamenco*-like piRNA clusters in these species (Figs 2A, S3A, and S3B). Importantly, the *gypsy-env* retrotransposons that we identified in these species have insertions in the clusters (indicated by colours in the figures), suggesting their role in suppressing them. We also found clusters that resemble germline clusters in *D. melanogaster*, in that they produce abundant piRNAs from both strands and have little bias in the orientation of *gypsy* insertions (Figs 2B and S3C). We performed fluorescent in situ hybridisation (FISH) using short-oligo DNA probes to examine the expression of the precursor transcripts from the clusters in the ovaries. FISH signals of the uni-stranded clusters were detected in the somatic cells, while the dual-stranded clusters were expressed in the germline cells in *D. pseudoobscura* and *D. bifasciata* (Figs 2C and 2D, and S3D–S3F). We further observed that these uni-stranded clusters have putative promoter peaks of RNA polymerase II and spliced precursor transcripts, resembling protein coding genes (S4A–S4C Fig). These are features common to the *flamenco* cluster in *D. melanogaster* [8]. We conclude that *D. pseudoobscura* and *D. bifasciata* have both somatic and germline piRNA clusters that resemble those of *D. melanogaster*.

We also found uni-stranded and dual-stranded piRNA clusters in *D. eugracilis* with similar arrangements of *gypsy* insertions (S5 Fig). We could not determine the cell type–specific expression of *D. eugracilis* clusters by FISH likely because they expressed much fewer piRNAs per kilo bases than clusters from the other species (compare S5 Fig with Figs 2 and S3). The uni-stranded cluster that we named 3031 produces piRNAs antisense to copies of *env*-carrying *Gypsy-6_Deu*, suggesting its role in silencing the element in the soma, and resides within an intron of a protein-coding gene (S5 Fig).

### Somatic biogenesis bodies are present in the *obscura* group species and *D. eugracilis*

piRNA biogenesis factors concentrate on cytoplasmic *flamenco* RNA to form Yb bodies in *D. melanogaster*. In the absence of Yb, other biogenesis factors, such as Armitage, disperse in the

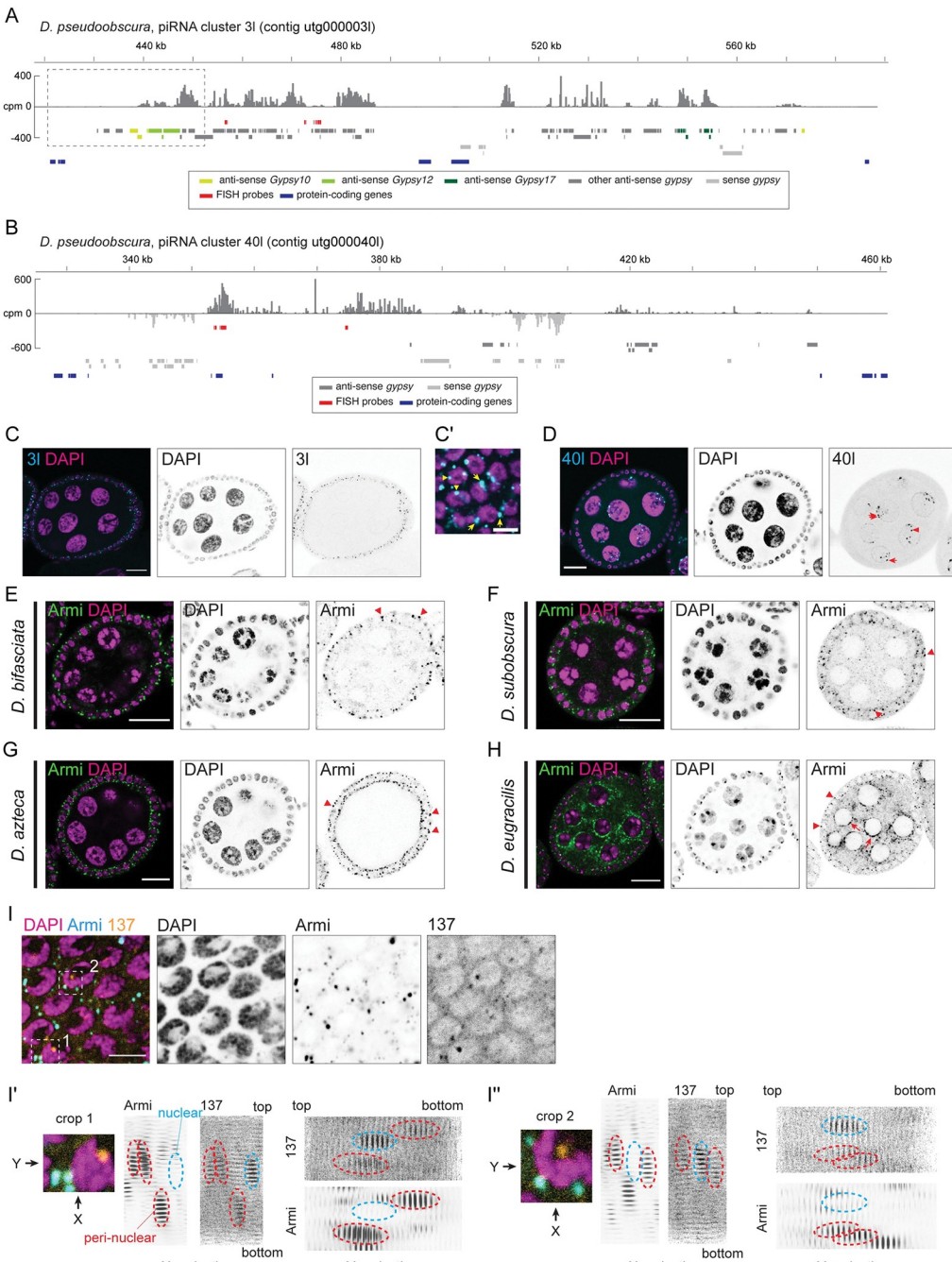

**Fig 2. Somatic and germline piRNA clusters in *D. pseudoobscura* ovaries resemble those of *D. melanogaster*. (A and B)** Shown is the coverage of piRNA reads (>22 nt) in counts per million genome mappers (CPM) from the oxidised whole ovary small RNA library of *D. pseudoobscura* that uniquely mapped to the cluster regions. Sense and antisense reads are coloured in dark and light gray, respectively. Coloured bars indicate *gypsy* insertions predicted by RepeatMasker, annotated protein-coding mRNA exons, and the FISH probes. A dotted box in (A) indicates the putative transcription start site of the cluster, for which a magnified view is shown in S4A Fig. **(C and D)** RNA FISH against transcripts from the piRNA clusters, showing the expression of the cluster 3l (C and C') and 40l (D) in the somatic and germline cells, respectively. A birds' eye view of the somatic epithelium of a FISH-staining of 3l is shown in (C'). Putative sites of transcription (in the nuclei) and processing (at the nuclear periphery) of the cluster transcripts are indicated by arrowheads, and arrows, respectively (C' and D). Scale bars = 20 μm in (C) and (D) and 5 μm in (C'). **(E–H)** Immunofluorescent staining of Armitage in *D. bifasciata* (E), *D. subobscura* (F), *D. azteca* (G), and *D. eugracilis* (H) egg chambers (Armitage in green and DAPI in magenta) showing the focused localisation in the somatic epithelium (arrowheads) and the perinuclear localisation in the germline (arrows). **(I)** Costaining of Armitage (blue) and the

transcript RNA from the cluster CM1_137 (orange) in a somatic epithelium of a *D. bifasciata* egg chamber. **(I') and (I")** Shown are 3D reconstructions of Z-stack images taken at regions indicated in (I). About 30 slices of confocal planes from the Armitage and the cluster CM1_137 stainings are aligned and projected from the X and Y axes to observe the colocalisation. Nuclear and perinuclear signals of CM1_137 are indicated by cyan and red dotted circles, respectively. Scale bars = 20 μm in (E–H) and 5 μm in (I).

cytoplasm and fail to produce abundant piRNAs from *flamenco* [27] (Fig 1A). The immuno-fluorescence staining in *D. bifasciata*, *D. subobscura*, *D. azteca*, and *D. eugracilis* ovaries showed focused localisation of Armitage in the somatic cells, resembling the *D. melanogaster* Yb body (Fig 2E-H). Furthermore, a costaining of Armitage and the transcripts from the uni-stranded cluster CM1_137 identified earlier in *D. bifasciata* ovaries revealed their colocalisation in the 3D-reconstructed images (Fig 2I), indicating the presence of the piRNA biogenesis bodies despite the absence of Yb. Notably, *armitage* gene has been duplicated in the *obscura* group genomes (e.g., LOC111076847 and LOC111074804 in *D. obscura* and more than 2 annotated copies in *D. miranda*). The apparent lack of staining in the germline of *D. azteca* could be due to a different and antibody-insensitive Armitage being expressed in the germline (Fig 2G). Armitage stained stronger in the germline of *D. eugracilis*, suggesting that the phasing mechanism is more prominent in the germline (Fig 2H; see below).

## Specialised somatic piRNA biogenesis for *gypsy* is evolutionarily conserved in *Drosophila*

The focal localisation of Armitage in the *obscura* group species and *D. eugracilis* indicates that they have still have mechanisms to efficiently process the cluster-derived transcripts into piR-NAs. To further characterise somatic piRNAs, we sequenced the embryonic (0 to 2 h postferti-lisation) small RNA pool, which is devoid of somatic material, and compared it to whole ovarian small RNAs.

We first confirmed that the method faithfully captures piRNAs from somatic and germline compartments. Firstly, piRNAs targeting known somatic retrotransposons were scarce in the embryonic pool in *D. melanogaster* [7,32] (S6A Fig). Furthermore, the piRNAs mapping to the somatic *flamenco* locus were enriched by more than 10-fold in the ovarian pool and were among the most abundant piRNAs in the soma. In contrast, the germline cluster 80F was equally repre-sented in both libraries (Fig 3B). We further observed that about three-quarters of piRNAs enriched in the ovaries come from discrete clusters like *flamenco*, or from other *gypsy* antisense insertions (Fig 3E). We finally estimated the abundance and composition of ovary-enriched somatic piRNAs in Yb-depleted *D. melanogaster* flies. We observed a reduction in *flamenco*-derived piRNAs by around 15-fold in the ovaries, while genic piRNAs were reduced by only 18% (Fig 3E). In contrast, the control RNAi ovaries expressed comparable levels of *flamenco*-derived piRNAs alongside other *gypsy* antisense piRNAs, which corroborate with previous findings [27].

The comparison of the embryonic and ovarian piRNA pools in the *obscura* group species and *D. eugracilis* mirrored the observations in *D. melanogaster*. Firstly, the transposons that abundantly produce somatic piRNAs are classified in the same subgroup as *gypsy*, indicating that somatic piRNAs in these species defend against similar types of transposons as *D. melano-gaster* [32] (S6B and S6C Fig). Secondly, piRNAs mapping to the somatic clusters identified earlier were enriched in the ovaries, while the germline clusters were equally represented in both libraries (Figs 3C and S6D). Additionally, we found that other uni-stranded clusters in *D. bifasciata* and *D. eugracilis* produce piRNAs in the soma. We further found that piRNAs from discrete clusters and from *gypsy* antisense insertions constitute a majority of somatic piRNAs (Fig 3E; 87.4%, 75.4%, and 72.5% for *D. pseudoobscura*, *D. bifasciata*, and *D. eugracilis*, respec-tively). The numbers for *D. bifasciata* and *D. eugracilis* were likely underestimated due to

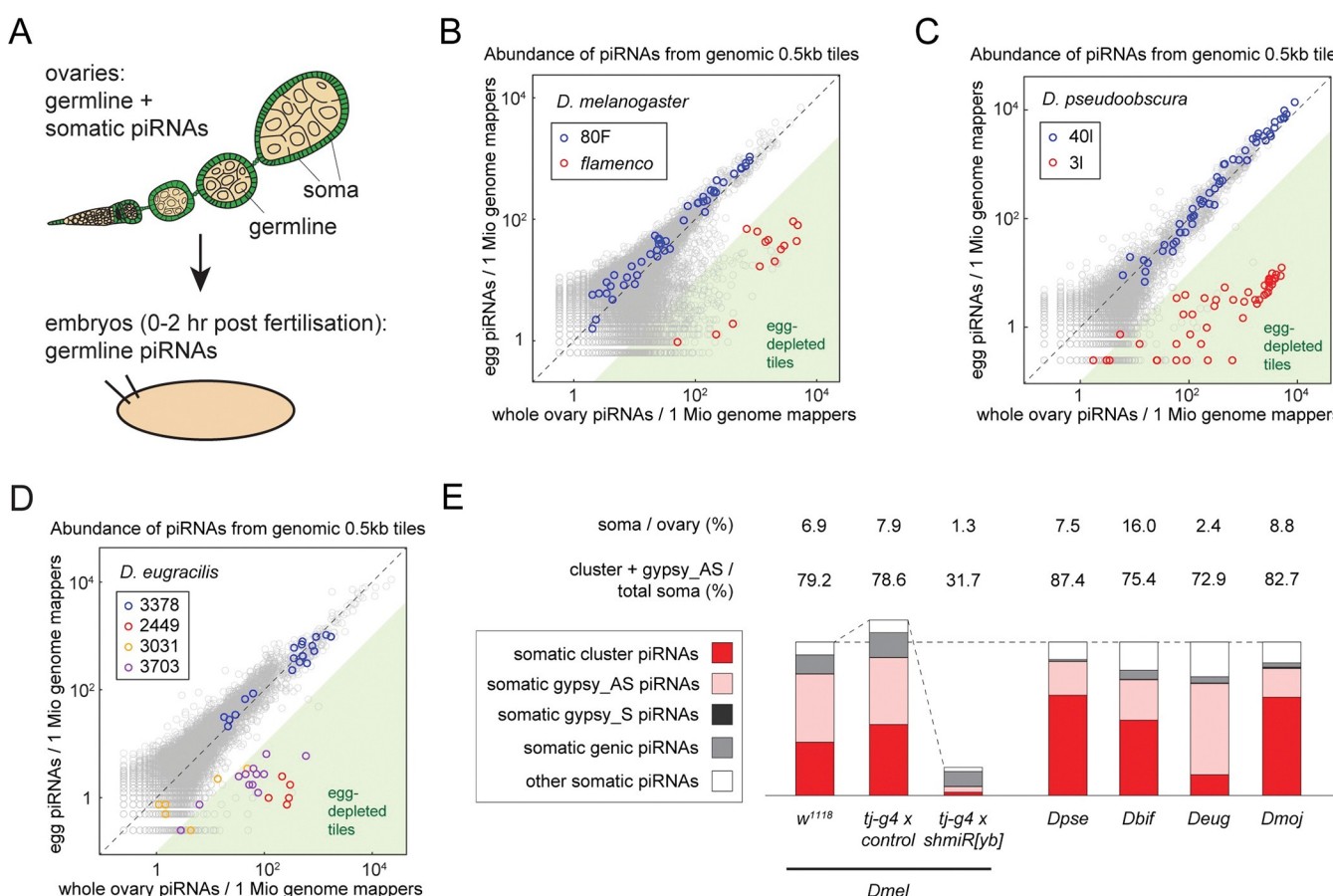

**Fig 3. Somatic piRNA biogenesis in *Drosophila* species is highly specialised for targeting *gypsy*.** **(A)** A schematic illustration contrasting the whole ovaries and the embryos that consist of both somatic (green) and germline (beige) cells and germline cells alone, respectively. Somatic piRNAs are only represented in the whole ovary small RNA pool while germline piRNAs are represented in both libraries. **(B–D)** Scatter plots showing the abundance of piRNAs from the whole ovaries (X axes) and the eggs (Y axes) that uniquely mapped to the individual 0.5 kb tiles of *D. melanogaster* (B), *D. pseudoobscura* (C), and *D. eugracilis* (D) genomes. Dual-stranded germline clusters are coloured in blue, while uni-stranded somatic clusters are coloured in orange, red, and purple. Tiles that expressed piRNAs more than 10 times in the whole ovaries than in the eggs are shaded in green. **(E)** Bar charts showing the abundance of different classes of somatic piRNAs from indicated genotypes and species. The estimated proportions of somatic piRNAs per ovarian piRNAs, the sum of cluster-derived piRNAs, and *gypsy* antisense piRNAs per total somatic piRNAs are shown. The bars are adjusted to the same height for the wild-type strains and are relative to the abundance of somatic piRNAs for the 3 strains of *D. melanogaster*. Species names are abbreviated as follows: *Dbif, D. bifasciata; Deug, D. eugracilis; Dmel, D. melanogaster; Dmoj, D. mojavensis; Dpse, D. pseudoobscura*. The underlying data can be found in S6 Data file.

relatively poor annotations of transposons in these genomes. These observations indicate that the somatic piRNA pathway in these species remains highly selective, despite the lack of Yb.

Similar observations were also made in *D. mojavensis*, an evolutionarily distant species from the *Drosophila/Drosophila* subfamily (Fig 1). We found a *flamenco*-like uni-strand piRNA cluster and confirmed that the cluster produces abundant piRNAs in the soma (S6 Fig). We also found that a majority of somatic piRNAs (82.7%) are either from discrete clusters or *gypsy* antisense insertions (Fig 3E). These observations indicate that specialised somatic piRNA biogenesis for *gypsy* appears to be a universal feature in the *Drosophila* taxa, regardless of the conservation of *yb*.

## Absence of ping-pong piRNAs in *D. eugracilis*

Ago3 is primarily required to bind transposon sense piRNAs to maintain the ping-pong amplification loop in *D. melanogaster*. Despite the absence of *Ago3* in addition to *Yb*, *D. eugracilis*

ovaries still produce piRNAs as abundantly as other species (Fig 4C). Strikingly, when we further examined these piRNAs, we found that most transposon piRNAs originate from the anti-sense strand. Of the 84 most piRNA-producing transposons in the *D. eugraclis* germline, 67 of them showed more than 95% bias towards antisense (Fig 4B). For example, the piRNAs mapping to *BEL-2_Deu* are almost exclusively antisense (Fig 4A). In contrast, similar transposons in *D. melanogaster* and *D. pseudoobscura* produce 15% to 40% of sense piRNAs (S7 Fig). The observed absence of sense piRNAs prompted an examination of ping-pong in *D. eugracilis*. We measured the distance between the 5′ ends of overlapping transposon sense and antisense piRNAs, where ping-pong pairs show a characteristic 10 nt overlap. Unexpectedly, of the 10 germline *D. eugraclis* transposons that produce at least 10% of sense and antisense piRNAs, none demonstrated any enrichment at the 10 nt overlap, highly suggestive of a lack of ping-pong in *D. eugracilis* (Figs 4D and S7C). In contrast, most germline transposons in *D. melanogaster* and *D. pseudoobscura* show strong tendencies of having the 10 nt overlap to the anti-sense piRNAs (Fig 4D, examples shown in S7 Fig).

## Piwi and Aubergine receive phasing piRNAs in *D. eugracilis*

Although *D. eugracilis* piRNAs lack ping-pong signatures, they retain other characteristics of piRNAs seen in *D. melanogaster*, such as their size and preference to start with a Uridine at the 5′ end (S8B Fig). During phased piRNA biogenesis, the endonuclease Zucchini/MitoPLD preferably cleaves in front of a Uridine simultaneously produce the 3′ end of the preceding and the 5′ end of the next piRNAs. Transposon piRNAs in *D. eugracilis*, such as those mapping to *BEL-2_Deu*, demonstrate characteristics associated with phasing including the characteristic head-to-tail arrangement of flanking piRNAs, as well as a Uridine bias immediately downstream of the 3′ end of the piRNAs (Fig 4E).

We further explored this preference to phasing biogenesis by performing an immunoprecipitation using specific antibodies for *D. eugracilis* Piwi and Aubergine, sequencing the small RNA pools and validating them against the embryonic small RNA sequencing (S8A and S8G Fig). Piwi, Aubergine, and Ago3 in *D. melanogaster* participate in ping-pong and phasing to varying extents and receive different pools of piRNAs. In contrast, we found that Aubergine and Piwi receive nearly identical populations of piRNAs in *D. eugracilis*, as seen in the size distribution, sense-antisense bias of transposon piRNAs, and at the level of individual piRNAs (S8C–S8F Fig). In addition, both proteins receive phasing piRNAs as indicated by the 3′-5′ linkage of the transposon antisense piRNAs (Figs 4F, S8E, and S8F). These observations strongly indicate that *D. eugracilis* predominantly rely on phasing for producing piRNAs in the germline.

## Phasing occurs without slicing by PIWI proteins in the *D. eugracilis* germline

Abundant phasing in the absence of ping-pong piRNAs in the *D. eugracilis* germline was not expected, as the slicing event by a PIWI RISC, as part of the ping-pong cycle, is required to trigger phasing from the RNA substrate in the *D. melanogaster* germline. On the other hand, slicing can occur without perfect sequence complementarity of the entire length of piRNA. Therefore, a piRNA produced from one genomic locus can initiate phasing from different loci *in-trans*, such as in mouse pachytene spermatocytes [23] (S9A Fig). To test whether germline piRNAs in *D. eugracilis* are produced in a similar manner, we devised a computational tool, which we named "*in-trans* ping-pong analysis" to measure the frequency of genome-wide slicing-triggered phasing events. Although the precise rule of piRNA target recognition is incompletely understood, the region between g2 and g10 positions is minimally expected to match

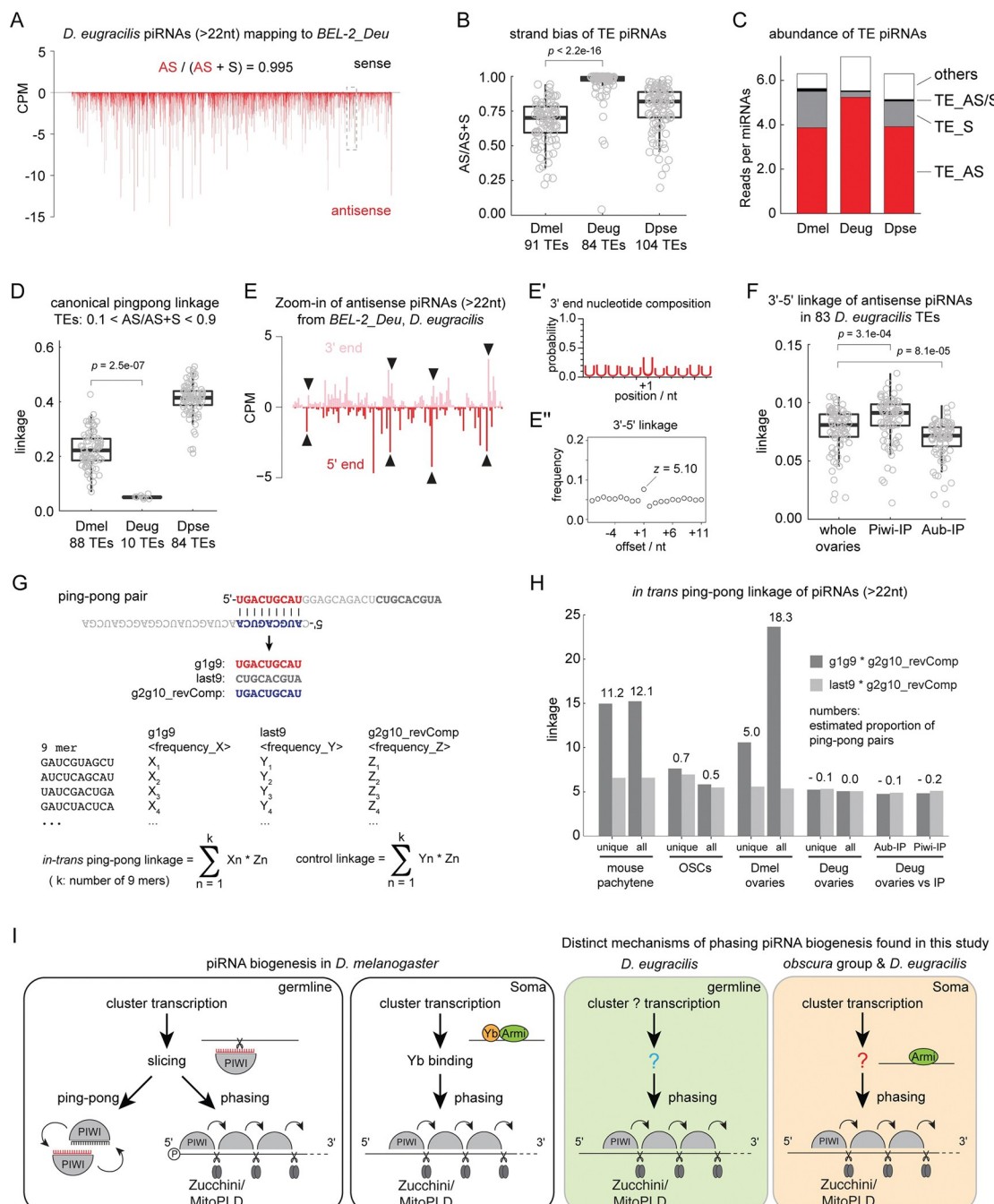

**Fig 4. Slicer-independent phasing predominantly produces transposon antisense piRNAs in *D. eugracilis* germline. (A)** Shown are the 5′ end coverage of piRNA reads (>22 nt) mapping to *BEL-2_Deu* from the oxidised whole ovary small RNA library of *D. eugracilis* in counts per million genome mappers (CPM). Sense and antisense reads are coloured in black and red, respectively. Antisense (AS) piRNAs make up more than 99% of all piRNAs mapping to *BEL-2_Deu*. **(B)** A box plot showing the strand bias of transposon-mapping piRNAs measured by the fraction of antisense reads out of the total reads. Circles represent individual transposons. Top approximately 100 transposons that produce most piRNAs in respective species are shown. *p*-Value is calculated by Mann–Whitney *U* test. **(C)** A bar chart showing the abundance of piRNAs of different classes relative to the abundance of microRNAs. Transposon piRNAs are grouped into sense (TE_S), antisense (TE_AS), and both (TE_AS/S). **(D)** A box plot showing the canonical ping-pong linkage of transposon-mapping piRNAs. Only the transposons that express at least 10% of both sense and antisense piRNAs are included. *p*-Value is calculated by Mann–Whitney *U* test. **(E)** Shown are the 5′ and 3′ ends of antisense piRNAs mapping to *BEL-2_Deu* from the dashed box in (A). piRNA 5′ ends are frequently found 1 nucleotide downstream of piRNA 3′ ends as indicated by arrowheads. **(E')** Shown are the frequencies of Uridines found at positions relative to the 3′ ends of piRNAs mapping to *BEL-2_Deu*. +1 corresponds to the immediate downstream nucleotide position. **(E'')** Shown is

the frequency plot of the 3′-5′ linkage of antisense *BEL-2_Deu* piRNAs. The z score of the linkage position +1 is shown (see methods). **(F)** A bar chart showing the 3′-5′ linkage values of transposon antisense piRNAs comparing the total ovarian piRNAs to Piwi- and Aubergine-bound piRNAs. *p*-Values are calculated by Mann–Whitney *U* test. **(G)** A summary of *in-trans* ping-pong analysis. 9mers from positions g1 to g9 (g1g9), last nine nucleotides (last9), and the reverse complemented g2 to g10 (g2g10_revComp) are extracted from all piRNA reads and frequencies of individual 9mer sequences are measured for each group. *in-trans* ping-pong linkage is calculated as the sum-product of frequencies between g1g9 and g2g10_revComp found in the same sequences while the sum-product between last9 and g2g10_recComp serves as a negative control. **(H)** Shown are *in-trans* ping-pong linkage values of genome unique piRNAs and all piRNA mappers per samples. Estimated proportions of *in-trans* ping-pong pairs out of all piRNAs are shown in percentage. "g2g10_revComp" of Aubergine- (Aub-) and Piwi-bound piRNAs were compared against "g1g9" and "last9" of the total ovarian piRNAs where indicated. **(I)** piRNA biogenesis mechanisms known in *D. melanogaster* and newly found in this study are summarised. In *D. melanogaster* ovaries, piRNAs are made by ping-pong and phasing where ping-pong is specific to the germline and phasing can be found in both germline and soma. Phasing is initiated by a slicing event in the germline, which also triggers the ping-pong loop, whereas Yb is required for recruiting Armitage (Armi) to the cluster-derived transcripts for phasing in the soma. Shown in colours are novel mechanisms of piRNA biogenesis found in this study. piRNAs are predominantly made by phasing without slicing events in the *D. eugracilis* germline, while Yb is dispensable for an efficient processing of cluster-derived transcripts in the somatic tissue of *obscura* group species and *D. eugracilis*. The underlying data can be found in the S6 Data file.

the target sequence [33,34]. We counted the frequencies of every 9mer found at the g1 to g9 part of piRNAs and measured how often they find identical sequences at the reverse-complemented g2 to g10 of other piRNAs (see methods and Fig 4G). Consistent with the previous findings [22,23], mouse pachytene-stage piRNAs showed a strong signature of *in-trans* ping-pong where more than 10% of piRNAs were estimated to have pairs (Fig 4H). As a negative control, less than 1% of piRNAs from the OSC lines (*D. melanogaster*), which is devoid of slicing-competent PIWI proteins, could find the ping-pong pairs. The *D. melanogaster* whole ovarian piRNAs showed strong *in-trans* ping-pong linkage, which also measures the abundance of canonical ping-pong pairs. The analysis estimated that there were fewer ping-pong pairs between uniquely mapped piRNAs (5.0%) than all piRNAs mappers (18.3%) in *D. melanogaster*. This is also seen in *D. pseudoobscura* piRNAs and consistent with the abundance of ping-pong piRNAs from transposons (Figs 4H and S9C). Strikingly, *in-trans* ping-pong pairs were not detected in *D. eugracilis* either in the pool of all genome mapping piRNAs or in the pool of genome-unique mappers (Figs 4H and S9C). An underrepresentation of Piwi- or Aubergine-bound piRNAs, which may take part in slicing-triggered phasing but make up a small fraction of the total piRNA pool, can be ruled out. This is because neither of them alone showed any linkage of *in-trans* ping-pong to the total pool of piRNAs (Fig 4H). We conclude that phasing piRNA biogenesis occurs independently of slicing in the germline of *D. eugracilis*.

## Discussion

We analysed *Drosophila* species that lack key known piRNA biogenesis factors in *D. melanogaster*. We demonstrated that the *obscura* group species and *D. eugracilis* are able to abundantly produce somatic *gypsy* antisense piRNAs in the absence of *yb*. We further demonstrated that *D. eugracilis* exclusively produces phasing piRNAs in the germline, independently of slicer activity.

### Somatic cluster Architecture is conserved across *Drosophila*

piRNA clusters are highly diverged in their genomic origins but retain general structural features over evolution [35–37]. A recent study showed that individual germline clusters are dispensable for silencing transposons in *D. melanogaster*, suggesting a rapid change in transposon contents and a slower turnover of cluster identities over evolution [38]. In contrast, the *flamenco* cluster in *D. melanogaster* is required for suppressing *gypsy* and maintaining female fertility. Here, we demonstrated that the *flamenco*-like cluster appears to be deeply

conserved in *Drosophila*, which we characterised by its strand bias, *gypsy* insertions, and somatic cell specificity. These clusters also carry copies of active *gypsy*, suggesting their functional importance. It is important to note that *env*-carrying *gypsy* elements are not only evolving within species, but also frequently transmitting between species likely due to their capacity to form virus-like particles [32]. It is therefore likely that the somatic piRNA clusters are under greater pressure than the germline clusters to adapt to rapidly changing *gypsy* elements.

## Evolutionary drivers of the loss of essential piRNA biogenesis factors

What drove the loss of *yb* and Ago3 considering their essential functions in *D. melanogaster* and the high degree of conservation across *Drosophila*? Different TEs produce different amounts of ping-pong and phased piRNAs in *D. melanogaster* [7]. The lack of phasing pathway in the germline leads to a nearly complete loss of piRNAs for some TEs while other TEs maintain piRNAs solely by ping-pong, suggesting that the balance between ping-pong and phasing is labile at the level of individual TEs [31]. The piRNA population adapts rapidly to changes in TE content [39]. Hence, it is possible that *Drosophila* species adjust their weight on ping-pong and phasing depending on the TEs that they have. Notably, ping-pong mechanism is very prominent for most germline TEs in *D. pseudoobscura* while no TEs in *D. eugracilis* rely on ping-pong (Fig 4D). These changes likely reflect on the history of TEs that they have been exposed to, which may have shaped the mode of piRNA biogenesis and eventually led to the loss of *yb* as well as *Ago3*.

## Gypsy antisense piRNAs can be produced independently of Yb

Despite the loss of *yb*, the *obscura* group and *D. eugracilis* are able to produce *gypsy*-antisense piRNAs to a high degree of specificity. The biogenesis factor Armi appears to form Yb body-like structures that colocalise with cluster-derived RNA in these species, which indicates that the downstream processing of piRNAs occurs in a similar manner as *D. melanogaster*. One possibility is that another protein has taken over the role of Yb by recruiting Armitage to the cluster transcripts. Since neofunctionalisation of duplicated gene paralogs is frequently found in the piRNA pathway [9,40,41], other Tudor proteins or DEAD-box RNA helicases may have once again duplicated in these species to replace Yb. Alternatively, the clusters in Yb-less species might be transcriptionally regulated in a way that transcripts are efficiently processed by Armitage in the absence of Yb. This idea seems consistent with the high ratio of cluster/TE piRNAs to genic piRNAs observed in this study (Fig 3H).

## Phased piRNAs can be made without ping-pong/slicer activity

A lack of Ago3 in *D. melanogaster* leads to the sense and antisense piRNAs being loaded onto Aubergine for a homotypic form of ping-pong [42,43]. Though sterile, the absence of Ago3 does not erase ping-pong signature in the ovaries. In contrast, Aubergine in *D. eugracilis* exclusively receives phased piRNAs, and no ping-pong signature was observed.

Although transposon defence mechanisms tend to evolve rapidly to defend against similarly rapidly evolving transposons, the ping-pong pathway is extremely highly conserved and is evolutionarily ancient [18,19,39]. An example of this is within Arthropods, in which biogenesis mode and the abundance of somatic piRNAs differ drastically between species, yet TE targeting piRNAs are still made through ping-pong in the germline [44]. The lack of Ago3 and ping-pong in *D. eugracilis*, therefore, may have much broader implications on the relationship between transposons and their silencing pathways.

Other species, such as *C. elegans*, have also diverged away from the standard piRNA pathway and rely on other biogenesis pathways to specify TE piRNAs [45]. Curiously, *D. eugracilis*

still appears to have much of the central mechanisms behind standard piRNA production, such as subcellular localisation and the 5′ Uridine preference of PIWI proteins. Furthermore, piRNA biogenesis factors that are known to be involved in ping-pong, such as Spindle-E [7], Vasa [46], and Qin [20], are all conserved in *D. eugracilis*. Although the precise mechanism is not known, these factors bind and stabilise PIWI proteins with precursor RNA at different steps in the ping-pong cycle [46,47]. Whether any of these factors participates in piRNA biogenesis in *D. eugracilis* and whether there is common logic in their molecular actions are both intriguing open questions.

The lack of ping-pong in *D. eugracilis* may have broader implications on the piRNA pathway as a whole, such as how germline piRNA clusters are transcribed and processed, let alone the driving force to lose such a robust and evolutionarily deeply conserved mechanism [18]. The Rhino-Cutoff-Deadlock (RDC) complex in *D. melanogaster* binds germline piRNA clusters, and couples transcription to nuclear export, then to piRNA biogenesis [9,48–50]. Most of the players in this circuit are again conserved at the gene level in *D. eugracilis*. However, this alone does not explain the extreme strand bias of transposon piRNAs in *D. eugracilis*, as Rhino-licenced transcription is bidirectional by nature utilising dispersed promoters within the clusters [40]. On the other hand, outside promoters also allow transcription through the cluster, potentially by suppressing splicing via the RDC complex [9,40,51]. Although the available genome assembly did not allow us to examine the possibility of outside promoters, *flamenco*-like clusters may exist in the *D. eugracilis* germline where transposons are aligned to the same direction and the strand bias of piRNAs is dictated by external promoters.

The lack of ping-pong, and the lack of piRNA-guided slicing as a whole, introduces another interesting conundrum. Phasing biogenesis requires a "triggering" event to initiate the endonucleolytic cleavage of piRNA precursors in the *D. melanogaster* germline (Fig 4I). The slicing activity of Ago3-bound and Aubergine-bound piRNAs is required for the production of nearly all germline phased piRNAs bound by Piwi [20,21]. It is possible that the *D. eugracilis* germline has become like the soma and has become capable of bringing precursor transcripts to Zucchini on the mitochondrial surface, without the need of endonucleolytic cleavage. However, this mode of piRNA biogenesis would be more promiscuous by nature in the absence of a "specificity" factor, such as Yb [27]. Therefore, *D. eugracilis* germline may have 2 distinct mechanisms to achieve specificity: the cluster definition and the selection of cluster-derived transcripts.

The logic of sequence complementarity of a piRNA and its target may go beyond ping-pong biogenesis. Maternally deposited Piwi- and Aubergine-bound piRNAs are responsible for silencing transposons in the next generation [52]. In transposon-induced hybrid dysgenesis, only the cross of a naïve father and a mother that carries loci that produce abundant transposon piRNAs can produce fertile progeny. It is thought that maternally inherited transposon piRNAs find the target RNA to produce more piRNAs from the same transposons. Therefore, the logic of sequence complementary is akin to the piRNA-dependent transgenerational silencing of transposons. We found that Aubergine and Piwi localise to the pole cells and other somatic nuclei in the *D. eugracilis* embryos before the zygotic transcription starts. Maternally inherited Aubergine and Piwi may serve the same purpose by other unknown mechanisms in *D. eugracilis*. Alternatively, they may have other functions, such as formation of the germ plasm [53], selectively degrading mRNAs in the germ plasm [54] or suppressing transposons in the somatic cells during embryonic development [55].

In summary, we found that the *obscura* group species and *D. eugracilis* have likely acquired novel mechanisms of phasing piRNA biogenesis distinct from those of *D. melanogaster* and other *Drosophila* species (summarised in Fig 4I). This raises mechanistic questions of what is possible in the piRNA pathway but also raises many biological conundrums to self and non-self-RNA distinction.

## Materials and methods

### Fly husbandry

We obtained wild-type strains of *D. pseudoobscura* (k-s12), *D. eugracilis* (E-18102), and *D. mojavensis* (k-s13) from the *Drosophila* species stock center at the Kyorin University, *D. bifasciata* (14012-0181-02), *D. subobscura* (14011-0131-04), and *D. azteca* (14012-0171-03) from The National Drosophila Species Stock Center at the Cornell University. The *D. melanogaster* strains, *w<sup>1118</sup>*, *armitage<sup>1</sup>*, and *armitage<sup>72.1</sup>* were obtained from the Bloomington Drosophila Stock Centre. *traffic-jam gal4* and shmiR lines for *yb* [31] and *armitage* [56] were obtained from Dr Dorothea Godt and Dr Julius Brennecke, respectively. Flies were raised at room temperature (approximately 22°C) in the standard food based on molasses, semolina, sugar, and fresh yeast.

### Identification of the *yb* homologues in *Drosophila* species

We examined the conservation of the *yb*, *boyb*, and *soyb* genes in a total of 356 genome assemblies of *Drosophila* species that were available in NCBI in February 2022 as well as 15 Nanopore genome assemblies published in Miller and colleagues [57]. We performed a tBlastn search using the default option of the stand-alone NCBI Blast package 2.9.0. The FlyBase entries FBpp0070462, FBpp0078210, and FBpp0292885 were used as the bait sequences for Yb, BoYb, and SoYb, respectively. The conservation of *yb* was called when the whole Hel-C domain (33–133 aa) and the DEAD-box RNA helicase domain (391–740 aa) found the homologous parts in a single locus. Additionally, the Hel-C domains of the Yb homologues from species that are distantly related to *D. melanogaster* were predicted by the HHpred (https://toolkit.tuebingen.mpg.de/tools/hhpred) and the whole protein sequences were used as baits for the tBlastn search. They include ALC48774.1 (*D. busckii*), XP_043072139.1 (*D. grimshawi*), XP_023179535 (*D. hydei*), XP_002055589.2 (*D. virilis*), XP_001965435 (*D. ananassae*), XP_017029052.1 (*D. kikkawai*), and XP_023034986.1 (*D. willistoni*). We also searched homologues of several other piRNA pathway genes in the same way. These include PIWI genes *piwi*, *aubergine*, and *Ago3*, and the core piRNA biogenesis factors *zucchini* and *vasa*. The results are summarised in S1 Data. We did not examine more than 2 genome assemblies from the same species when the *yb* homologues were identified in 2 independent assemblies.

### Identification of intact *env*-carrying *gypsy* insertions in *Drosophila* genomes

We ran RepeatMasker 4.1.0 to predict the *gypsy* insertions in the following genome assemblies: the UCI_Dpse_MV25 assembly (RefSeq accession: GCF_009870125.1) and the Nanopore assembly of the *D. pseudoobscura* genome (PMID: 30087105), the UCBerk_Dbif_1.0 assembly of the *D. bifasciata* genome, the DaztRS1 assembly (GenBank accession: GCA_005876895.1) of the *D. azteca* genome, and the ASM1815383v1assembly of the *D. eugracilis* genome. We used the RepBaseRepeatMaskerEdition-20181026 for finding transposable elements. Separately, we searched for the genomic loci that contain all open reading frames of the *env*-carrying *gypsy* retrotransposons using the sequences obtained from the RepBase. We searched copies of *Gypsy10_DPse*, *Gypsy17_Dpse*, and *Gypsy12_Dpse* in the *D. pseudoobscura* genome, *Gypsy17_Dpse* and *Gypsy_DS* in the *D. bifasciata* genome, *Gypsy-3_DAzt*, *Gypsy-8_DAzt*, *Gypsy-19_DAzt*, and *Gypsy-101_DAzt* in the *D. azteca* genome, and *Gypsy1_DM*, *Gypsy-6_DEu* and *Gypsy-37_DEl* in the *D. eugracilis* genome. We called RepeatMasker-predicted *env*-carrying *gypsy* insertions as "intact" when all ORFs were present. The genomic loci of all identified copies and the detailed criteria of calling the intact insertions can be found in S3 Data.

## Immunofluorescence staining of ovaries

Rabbit polyclonal antibodies were generated by Genscript using the following peptides where additional single Cysteine residues for the cross-linking purpose are indicated in lowercase: *D. pseudoobscura* Piwi (MSENQGRGHRRPHGc), Aubergine (MNDLPTNSGHSRGRc), and Ago3 (MSGRGNLLKLFNKKc), *D. eugracilis* Piwi (GRRRPLYDEEPSTSc) and Aubergine (cANKQGDPRGPVSGR). Purified IgG were reconstituted in PBS (0.5 μg/ml) before use. All five antibodies predominantly detected single bands at expected sizes (100 approximately 110 kDa) by western blotting (S2A and S2B Fig). Mouse monoclonal anti *D. melanogaster* Armitage (1D1-3H10), which was raised against the peptide from 36 to 230 aa of FBpp0100102, is a gift from Julius Brennecke. The ovaries were freshly dissected from 2- to 7-d-old females in PBS and fixed in PBS containing 4% formaldehyde for 10 min at room temperature. Fixed ovaries were permeabilised in PBS containing 0.5% v/v Triton-X for 30 min, washed in PBS containing 0.1% Triton-X (PBS-Tx) several times before blocking in PBS-Tx containing 0.05% w/v BSA for 30 min. The primary antibody incubation was conducted in the blocking solution at 4˚C overnight with antibodies in the following dilutions: 1:500 for anti *D. pseudoobscura* Piwi, Aubergine, and Ago3, anti *D-eugracilis* Piwi and Aubergine, and 1:200 for anti *D. melanogaster* Armitage. Goat anti-rabbit or mouse IgG conjugated to Alexa Fluor 488 or 568 (Abcam) were used as secondary antibodies, and the confocal images were taken on a Zeiss LSM-800. DAPI was used to visualise DNA. Images were processed by Fiji.

## Immunofluorescence staining of embryos

*D. eugracilis* embryos of 1 to 1.5 h postfertilisation were collected, and the chorions were removed by bleaching. Embryos were then fixed in heptane saturated by formalin for 30 min at room temperature and washed by methanol to remove the vitelline membrane and the residual heptane. Subsequently, embryos were fixed again by PBS containing 4% formaldehyde for 10 min at room temperature before the permeabilization and the antibody incubations as described for the ovaries.

## Western blotting

Around 10 to 20 μl of freshly dissected ovaries was collected in cold PBS and snap-frozen. The ovaries were homogenised in a 10-times volume of the RIPA buffer (50 mM Tris–HCl (pH 7.5), 150 mM NaCl, 1% TritonX-100, 0.1% SDS, 0.1% Na-deoxycholate, 1 mM EDTA, and 1× cOmplete protease Inhibitors (Roche)) on ice. The lysate was cleared by centrifugation, boiled in Laemmli buffer before loading onto the SDS-PAGE. The gel was transferred onto a Nitro-cellulose membrane and incubated with the primary antibodies in the following dilutions: 1:500 for anti *D. pseudoobscura* Piwi, Aubergine, and Ago3, anti *D-eugracilis* Piwi and Aubergine. The HPR-conjugated secondary antibodies were used for the standard ECL detection.

## RNA in situ fluorescent hybridisation (FISH)

RNA FISH protocol was modified from Andersen and colleagues [40]. Briefly, short oligo DNAs for each target (see S5 Data) were pooled and labelled with 5-Propargylamino-ddUTP-Cy3 or -Cy5 (Jena Biosciences) using the Terminal Deoxynucleotidyl Transferase (Thermo Fisher) [58], which yielded at least 50% labelling efficiencies. Ovaries were freshly dissected from 2- to 7-d-old females and fixed in PBS containing 4% formaldehyde. The fixed ovaries were permeabilised overnight at 4˚C in 70% ethanol and washed twice by RNA FISH wash buffer (10% (v/w) formamide in 2× SSC). Subsequently, the ovaries were resuspended in 50 μL Hybridization Buffer (10% (v/w) dextran sulfate and 10% (v/w) formamide in 2× SSC)

and incubated with 1 pmol of labelled oligo probes for overnight at 37˚C. The ovaries were then washed several times in RNA FISH Wash Buffer and stained by DAPI before mounting. For the double-staining of FISH probes and antibodies, we first performed the immunostaining and followed the same procedure of the FISH from the permeabilization step. Confocal images were taken on a Zeiss LSM 800 and processed by Fiji.

### RT-qPCR analysis of *gypsy* expression

Freshly dissected ovaries from 2- to 7-d-old females were homogenised in Trizol to obtain the total RNA. The total RNA was further treated with RNase-free DNase I (NEB, M0303) and reverse-transcribed using the random hexamer and Superscript II (Invitrogen/Thermo Fisher) following the manufacturer's protocol. qPCR was performed using GoTaq DNA polymerase (Promega), EvaGreen (Biotium), and the following primers: *rp49*-fw: CCGCTTCAAGGGA CAGTATCTG, *rp49*-rv: ATCTCGCCGCAGTAAACGC, *gypsy*_spliced-fw: CAACAATCT GAACCCACCAATCT, *gypsy*_spliced-rv: TATGAACATCATGAGGGTGAACG. The level of *gypsy* expression was normalised to *rp49*. The expression of *gypsy* in the wild-type ovaries was below detection, lower than a second control ovaries that weakly overexpressed *gypsy*. The level of *gypsy* expression in the Yb-depleted ovaries was thus compared to the second control in Fig 1B.

### RNA sequencing

Polyadenylated RNA was purified from the DNase-treated total RNA using the oligo d(T)25 magnetic beads (NEB, S1419) and used for the library preparation. Libraries were cloned using the NEBNext Ultra Directional II RNA Library Prep Kit for Illumina (NEB, E7760), following the manufacturer's instruction, and amplified by the KAPA LongRange DNA polymerase (Sigma, KK3502) using the universal forward primer, Solexa_PCR-fw: (5′-AATGATACGGC GACCACCGAGATCTACACTCTTTCCCTACACGACGCTCTTCCGATCT) and the barcode-containing reverse primer TruSeq_IDX: (5′-CAAGCAGAAGACGGCATACGAGAT xxxxxxGTGACTGGAGTTCAGACGTGTGCTCTTCCGATCT where xxxxxx is the reverse-complemented barcode sequence). QuantSeq 3′ mRNA-Seq Library Prep Kit FWD for Illumina from Lexogen was used to generate the mRNA 3′-end sequencing library from the total RNA of *D. eugracilis*. Amplified libraries were multiplexed and sequenced on a HiSeq platform in the paired-end 150 bp mode by GENEWIZ/Azenta.

### RNA sequencing analysis

Both R1 and R2 reads from the poly-A+ RNA sequencing reads, and the R1 reads from the mRNA 3′-end sequencing were trimmed of the Illumina-adapter sequences using the FAS-TX-Toolkit from the Hannon Lab. An additional stretch of poly-A sequences was also removed from the mRNA 3′-end sequencing reads. The trimmed reads were subsequently filtered by the sequencing quality. Only the paired and unfiltered reads were then mapped to the Nanopore assembly (Miller and colleagues [57] (PMID: 30087105)) of the *D. pseuduoobscura* genome, the UCBerk_Dbif_1.0 assembly (GenBank accession: GCA_009664405.1) of the *D. bifasciata* genome, and the ASM1815383v1assembly (RefSeq accession: GCF_018153835.1) of the *D. eugracilis* genome using STAR 2.7.3a allowing up to 3 mismatches. The coverage of uniquely mapped reads was counted using bedtools 2.28.0 and normalised to 1 million genome-unique mappers.

## Chromatin immunoprecipitation (ChIP) sequencing

ChIP was performed as previously described [40]. Briefly, approximately 100 μl of ovaries were dissected into ice-cold PBS and cross-linked PBS containing 2% paraformaldehyde for 10 min at room temperature while mixing several times. The cross-linking was quenched by Glycine. The cross-linked ovaries were washed in PBS twice and sonicated in 1.2 ml of the ChIP lysis buffer (150 mM NaCl, 20 mM Tric-Cl (pH 8.0), 2 mM EDTA, 1% Triton-X and 0.1% SDS) using the BioRuptor (10 cycles of 30 s/30 s ON/OFF). The lysate was cleared and incubated overnight at 4°C with 1 μg of mouse monoclonal anti RNA polymerase II CTD (gift from Hiroshi Kimura [59]). Antibodies were captured by 20 μl of a 50:50 mix of Protein-A and Protein-G Dynabeads (Life Technologies). The beads were washed several times sequentially in the buffers containing low salt (150 mM NaCl), high salt (500 mM NaCl), 250 mM LiCl, and the TE buffer before eluting in the Elution buffer (1% SDS, 0.1 M NaHCO3, and 10 mM DTT). The eluate was treated with RNase Cocktail (Thermo Fisher, AM2286), de-crosslinked for 5 h at 65°C. De-crosslinked samples were treated with Proteinase K, and the DNA was extracted by phenol: chloroform and precipitated by 2-propanol. The DNA was then End-repaired and ligated to the NEBNext adaptor for Illumina (NEB, E7337A). The libraries were amplified by KAPA polymerase using the same primers as for the RNA sequencing.

## ChIP sequencing analysis

ChIP sequencing reads were trimmed of the Illumina-adapter sequences and filtered by quality using the FASTX-Toolkit. Read 1 reads were then mapped to the *D. pseudoobscura* Nanopore assembly and the UCBerk_Dbif_1.0 assembly using bowtie 1.2.3 allowing up to 3 mismatches. The coverage of uniquely mapped reads was counted using bedtools 2.28.0 and normalised to 1 million genome-unique mappers.

## Small RNA cloning

We generated small RNA libraries from 5 μg of oxidised or unoxidised total RNA using a modified protocol of the original method [60]. To prepare oxidised small RNA libraries, the size of 19 to 35 nt of RNA was first selected from the total RNA by PAGE using radio-labelled 19mer spike (5′-CGUACGCGGGUUUAAACGA) and 35mer spike (5′-CUCAUCUUGGUCGUA CGCGGAAUAGUUUAAACUGU). The size-selected RNA was precipitated, oxidised by sodium periodate [61], and size-selected for the second time by PAGE. To prepare unoxidised small RNA libraries, 2S ribosomal RNA was first removed from the total RNA using a biotinylated oligo DNA from IDT (5Biotin-TEG/TACAACCCTCAACCATATGTAGTCCAAGCA), followed by the size-selection using 19mer and 35mer spikes. The size-selected small RNAs were ligated to the 3′ adapter from IDT (5rApp/NNNNAGATCGGAAGAGCACACGTCT/ 3ddC where Ns are randomised) using the truncated T4 RNA Ligase 2, K227Q (NEB), followed by a third and second PAGE for oxidised and unoxidised libraries, respectively. Subsequently, the RNA was ligated to the 5′ adaptor from IDT (ACACUCUUUCCCUACACGAC GCUCUUCCGAUCUNNNN where Ns are randomised) using the T4 RNA Ligase 1 (NEB). Adaptor-ligated RNA was reverse-transcribed by SuperScript II and amplified by KAPA polymerase using the same primers as for the RNA sequencing.

## Preparation of embryonic small RNA libraries

Embryonic small RNA libraries were prepared from the total RNA collected from 0- to 2-h-old eggs laid by females that were reared at room temperature in a cage with apple agar plates. Embryos were bleached and immediately transferred to TriZol before preparing the total

RNA, which was subsequently treated with the DNase. The size-selected RNA was oxidised before ligating the adaptors.

## Immunoprecipitation of *Drosophila eugracilis* Piwi and Aubergine for small RNA cloning

Approximately 100 μl of *D. eugracilis* ovaries were dissected into PBS on ice. The tissue was homogenised on ice twice in 0.3 ml of the RIPA buffer. The lysate was cleared by centrifugation and diluted with 2.4 ml of IP dilution buffer (50 mM Tris–HCl (pH 7.5), 150 mM NaCl) and split into 2 reactions. Around 5 μg of antibodies against Piwi and Aubergine were each coupled to 50:50 of protein-A and protein-G Dynabeads (Life Technologies). Lysates were incubated with bead-coupled antibodies, rotating at 4˚C overnight. Subsequently, the beads were captured and washed 5 times with IP wash buffer (50 mM Tris–HCl (pH 7.5), 500 mM NaCl, 2 mM MgCl2, 10% glycerol, 1% Empigen). For Aub IP, 150 mM NaCl was used instead of 500 mM NaCl. The IP was eluted in 10mM DTT and 0.1% SDS in 1× TE at 85 degrees. The bound RNA was extracted using acid-phenol:chloroform followed by 2-propanol precipitation, mixed with the radio-labelled spikes before the size selection. The size-selected RNA was oxidised before ligating the adaptors.

## Analysis of *Drosophila* small RNA sequencing libraries

The R1 sequencing reads were trimmed of the Illumina-adapter sequence using the FAS-TX-Toolkit. The 4 random nucleotides at both ends of the read were further removed. The trimmed reads of 18 to 40 nt in size were first mapped to the infrastructural RNAs, including ribosomal RNAs, small nucleolar RNAs, small nuclear RNAs, microRNAs, and transfer RNAs (tRNAs) using bowtie 1.2.3 allowing up to one mismatch. Sequences annotated in the dm6 r6.31 assembly of the *D. melanogaster* genome were used. microRNA reads were used for a normalisation purpose. Reads that originate from the 19mer and 35mer spikes were also removed. The remaining reads were used for all the downstream analyses. The trimmed and unfiltered reads were then mapped to the Nanopore assembly of the *D. pseuduoobscura* genome, the UCBerk_D-bif_1.0 assembly of the *D. bifasciata* genome, and the ASM1815383v1assembly of the *D. eugracilis* genome, the ASM1815372v1 assembly (RefSeq accession: GCF_018153725.1) of the *D. mojavensis* genome, and the dm6 r6.31 of the *D. melanogaster* genome using bowtie allowing up to one mismatch. Reads that mapped to the 100 nt upstream and downstream genomic regions of tRNA insertions were also removed. The method of identifying tRNA gene loci in the unannotated genome is further described in the code available in the git repository. Bedtools was used to count the coverage of the mapped reads. The genome-unique mappers from the oxidised total small RNA libraries were used to visualise piRNAs expressed from the piRNA clusters.

The small RNA library (SRR1746887) of the *D. melanogaster* OSCs from the previous study [13] was analysed in the same manner as other libraries generated in this study.

For the mapping of transposon-derived small RNAs, we first selected reads that mapped at least once to the respective genomic sequences. Genome-mapped unfiltered reads were subsequently mapped to the collection of autonomous *Drosophila* transposon sequences retrieved from the RepBase in March 2022, or to the curated set of transposon sequences available for *D. melanogaster* [21], using bowtie allowing up to 3 mismatches with the "—all—best—strata" option. We selected transposons in the *D. pseudoobscura* and *D. eugracilis* genomes for measuring the strand bias of piRNAs and the linkage analyses using the following criteria. We started with 150 RepBase entries that expressed most abundant piRNAs in the whole ovary samples of each species. Following, entries of similar sequences were removed by counting the piRNA reads that were mapped to multiple entries. This process resulted in 109 and 94

"nonredundant" transposon entries from *D. pseudoobscura* and *D. eugracilis*, respectively. For *D. melanogaster*, we started with 98 curated transposons that produced most abundant ovarian piRNAs in the $w^{1118}$ strain. Additionally, *mariner* transposons in *D. eugracilis* were excluded from the analysis of the piRNA strand bias because of the terminal inverted repeats. All the transposon sequences used in this study are available in the git repository.

We used the replicate 1 of the oxidised whole ovary small RNA library from *D. eugracilis* for all the analyses and additionally used the replicate 2 for the *in-trans* ping-pong analysis.

## Tile coverage analysis of the small RNA sequencing reads

We carried out 2 genomic tile analyses to measure the abundance of piRNAs. We only considered reads that are 23 nt or longer as piRNAs. We used the genome-unique mappers and uniquely mappable 0.5 kb tiles to generate the scatter plots, and all genome mappers and 0.2 kb tiles to quantify the soma-enriched piRNAs. We first identified 0.5 kb tiles that are covered at least 85% by unique regions by mapping artificially made 25mers against the whole genome. We then counted the number of reads that mapped to individual uniquely mappable 0.5 kb tiles. Counts were normalised to the number of genome-unique piRNA mappers. To count the abundance of all genome piRNA mappers in the 0.2 kb tiles, we first mapped the sequencing reads to the genome using bowtie with the "—all—best—strata" option and divided individual mapping instances by the number of mapping events per read, in order to evenly distribute multimappers across all repeats in the genome. We then counted the number of reads that mapped to individual 0.2 kb tiles. "Somatic" 0.2 kb tiles were defined as those tiles that expressed more than 10 times piRNAs in the whole ovaries than in the embryos. Of the somatic tiles, we defined "*gypsy*" sense and antisense tiles as those tiles that are covered at least half by the *gypsy* insertions predicted by the RepeatMasker. We applied the same rule to annotate the "mRNA exon" tiles and the piRNA cluster tiles. We used the RefSeq annotations for *D. pseudoobscura*, *D. eugracilis*, and *D. mojavensis*, and the FlyBase dm6 r6.31 annotations for *D. melanogaster*. We used the RNA sequencing read coverage of greater than 1 read per kilo base per million reads (RPKM) to define mRNA exons in the *D. bifasciata* genome. The coordinates of the piRNA clusters used in this study can be found in the git repository. When multiple annotations are found in the same tile, we chose the annotations in the order of "piRNA cluster," "*gypsy* antisense/sense," and "mRNA exons."

## Linkage analysis of ping-pong and phasing piRNA biogenesis

We carried out the linkage analysis of transposon piRNAs as previously described [62]. Briefly, 5′ and 3′ ends of piRNA mappers (23mer or longer) were counted at each transposon coordinates. Frequencies of the co-occurrence at the linkage positions were measured across the transposon length and weighed by the abundance of piRNAs. The linkage positions of +10 between 5′ ends of sense and antisense piRNAs, and +1 between the 3′ ends and the 5′ ends of antisense piRNAs were measured for the ping-pong and the phasing biogenesis, respectively. Frequencies were measured in the window of 20 nt and the Z scores are calculated as the deviation of the frequency value at the linkage position from the mean frequency divided by the standard deviation of the frequencies. Transposons that expressed at least 10% from both strands were included, and those that expressed more than 3 times in the whole ovaries compared to the embryos were excluded from the ping-pong linkage analysis.

## Analysis of mouse small RNA sequencing libraries

The small RNA library of mouse pachytene stage spermatocytes from the previous study (SRR1104823) was analysed [63] in the same way as *Drosophila* small RNA libraries. Briefly,

sequencing reads were trimmed of the adaptors. Reads mapping to the infrastructural RNAs were filtered. Sequences of the mouse infrastructural RNAs were retrieved from the Ensembl GRCm39 assembly and the RepBase. Followingly, 18 to 40 nt long reads were mapped to the GRCm39 genome using bowtie allowing up to one mismatch. Reads that are 23 nt in size or longer were considered as piRNAs and used for the *in-trans* ping-pong analysis.

## Linkage analysis of *in-trans* ping-pong

Individual genome-mapping piRNA reads were trimmed to the following 3 regions: the g1 to g9 position (g1g9) and the most 3′ 9 nucleotides (last9) of the sense piRNA read; and the reverse-complemented sequence from the g2 to g10 position (g2g10_revComp) where the g1 is the 5′ end of a piRNA and the g2 is the penultimate 5′ end, and so on. The frequencies of every different 9mers from each category were counted using jellyfish 2.3.0 and normalised to 1 million genome-mappers. The sum of the frequencies was arbitrarily set to 1,000. Products of the frequencies from the g1g9 and the g2g10_revComp were calculated for each 9mer and summed up to yield the *in-trans* ping-pong linkage value. The linkage value increases when 2 piRNAs form an *in-trans* ping-pong pair and the same sequence occurs at higher frequencies both in the g1g9 and the g2g10_revComp. We do not expect to observe any linkage between the last9 and the g2g10_revComp; therefore, the products between them were considered a genomic background. To estimate the proportion of the *in-trans* ping-pong pairs in the piRNA pool, we performed a simulation where we artificially added fixed number of pairs into randomly selected pool of 9mers. The artificial ping-pong pairs were set to have 100 counts per million reads (CPM). piRNA reads that are equal to or more abundant than 100 CPM make up about 10% of the whole *Drosophila* ovarian piRNA population. The simulation after including 0%, 1%, 3%, 5%, and 10% of artificial ping-pong pairs showed that the *in-trans* ping-pong linkage value increases by 1 as the proportion of the ping-pong pairs increases by 1% (S9B Fig). Based on this result, we estimated the percentage of *in-trans* ping-pong pairs in biological samples by the linkage value of g1g9 * g2g10_revComp subtracted by the value of last9 * g2g10_revComp.

## Supporting information

**S1 Fig. The conservation and expression of *yb* gene in *D. eugracilis*, *D. willistoni*, and *D. virilis*. (A)** Both poly-A+ RNA-seq and 3′ mRNA-seq show no expression of the pseudogene of *yb* in *D. eugracilis*. The coverage of neighbouring genes is shown in counts per million reads (CPM). Single and double asterisks indicate peaks of the 3′ mRNA-seq that correspond to canonical mRNA 3′ ends and a suspected internal priming, respectively. **(B and C)** Shown are the syntenic loci of *yb* gene in *D. willistoni* (B) and *D. virilis* (C) genomes. Genes that are found near the *yb* gene both in *D. melanogaster* and *D. willistoni* or *D. virilis* are coloured in blue, while genes that are found near the *yb* gene only in *D. willistoni* or *D. virilis* are coloured in gray.
(PDF)

**S2 Fig. PIWI protein localisation in the egg chambers of the *obscura* group ovaries and the blastderm stage of the *D. eugracilis* embryos. (A and B)** Antibodies raised against peptides from *D. pseudoobscura* and *D. eugracilis* PIWI proteins were used in (A) and (B), respectively. The lysates were prepared from ovaries from indicated species and blotted for antibodies against Piwi, Aubergine (Aub), and Ago3. Dominant bands, as indicated by red dots, were detected at predicted sizes (100 to 110 kDa) in all cases. Species names are abbreviated as follows: *Dazt, D. azteca; Dbif, D. bifasciata; Deug, D. eugracilis; Dpse, D. pseudoobscura; Dsub. D.*

subobscura. (C) Immunofluorescent stainings of (Ago3 in green and DAPI in magenta) of *D. azteca*, *D. bifasciata*, and *D. subobscura* egg chambers show a perinuclear localisation of Ago3 in all 3 species. Perinuclear granules, as indicated by arrowheads, are seen in all species. (D) Immunofluorescent staining of Piwi and Aubergine in *D. eugracilis* mid-blastoderm stage embryos showing the localisation of Piwi and Aubergine in the pole cells (marked by arrowheads) and in the zygotic somatic nuclei (Piwi only). Embryos of 1–1.5 h postfertilisation were stained. This is the stage after the nuclei reach the periphery of the blastoderm before the nuclear elongation and the onset of cellularisation. The bulk of zygotic transcription has not started in this stage. Scale bars = 20 μm. The underlying data can be found in S1 Raw Images file.
(PDF)

**S3 Fig. piRNA clusters in *D. bifasciata* ovaries. (A–C)** Shown are the coverage of piRNA reads (>22 nt) in counts per million genome mappers (CPM) from the oxidised whole ovary small RNA library of *D. bifasciata* that uniquely mapped to the cluster regions. Sense and antisense reads are coloured in dark and light gray, respectively. Coloured bars indicate *gypsy* insertions predicted by RepeatMasker, annotated protein-coding mRNA exons, and the FISH probes. Dotted box in (A) and (B) indicate the putative transcription start sites of the cluster, for which magnified views are shown in S4B and S4C Fig. **(D–F)** RNA FISH against transcripts from the piRNA clusters. Somatic signals as indicated by open arrows are seen for the FISH against clusters CM1_137 (D) and CM1_99 (E), while the FISH against the cluster CM1_135 (F) only stains the germline cells. The FISH against CM1_137 weakly stained the germline at the nuclear periphery (asterisk), while distinct perinuclear puncta are seen in the FISH against CM1_99 and CM1_135 (arrows). Putative sites of transcription in the nuclei as indicated by arrowheads are only seen in CM1_135.
(PDF)

**S4 Fig. Somatic piRNA clusters in the *obscura* group resemble protein-coding genes. (A–C)** Shown are the coverage of RNA polymerase II ChIP-seq (blue), and its input controls (light blue), poly-A+ RNA-seq (sense in red and antisense in pink), and the piRNA reads (>22 nt) at the 5′ end of the somatic piRNA clusters. Y-axes indicate counts per million genome mappers (CPM) for all tracks. RNA-seq read pairs that span introns are indicated as red and gray bars in pairs. Coloured bars at the bottom indicate *gypsy* insertions and annotated protein-coding mRNA exons. Peaks of RNA polymerase II in front of the clusters and at the neighbouring gene promoter regions are marked by single and double asterisks, respectively. Triple asterisks indicate protein-coding exons in *D. bifasciata* predicted by homology to *D. pseudoobscura* proteins.
(PDF)

**S5 Fig. piRNA clusters in the *D. eugracilis* genome. (A–D)** Shown are the coverage of piRNA reads (>22 nt) in counts per million genome mappers (CPM) from the oxidised whole ovary small RNA library of *D. eugracilis* that uniquely mapped to the cluster regions. Sense and antisense reads are coloured in dark and light gray, respectively. Coloured bars indicate *gypsy* insertions predicted by RepeatMasker, and an annotated protein-coding mRNA exon. The entirety of the uni-stranded clusters could not be determined because they are found at the end (indicated by double dashed lines) of the chromosome contigs.
(PDF)

**S6 Fig. Characterisation of somatic piRNA expression in *Drosophila* species. (A–C)** Abundance of ovarian (X axes) and embryonic (Y axes) piRNAs mapping to individual transposons in *D. melanogaster* (A), *D. pseudoobscura* (B), and *D. eugracilis* (C) genomes. Transposons that

expressed piRNAs more than 3 times in the ovaries than in the embryos are marked. Colours indicate families within the "errantiviridae/412" group of Ty3/Gypsy superfamily: red; group "Gypsy," green; group "17.6," cyan; group "412/mdg1," and black; unclassified. **(D and E)** Scatter plots showing the abundance of piRNAs from the whole ovaries (X axis) and the eggs (Y axis) that uniquely mapped to the individual 0.5 kb tiles of the *D. bifasciata* genome in (D) and *D. mojavensis* genome in (E). The dual-stranded germline clusters are coloured in blue while uni-stranded somatic clusters are coloured in orange and red. Tiles that expressed piRNAs more than 10 times in the whole ovaries than in the embryos are shaded in green. **(F)** Shown is the coverage of piRNA reads (>22 nt) in counts per million genome mappers (CPM) from the oxidised whole ovary small RNA library of *D. mojavensis* that uniquely mapped to the cluster region. Sense and antisense reads are coloured in dark and light gray, respectively. Coloured bars indicate sense and antisense *gypsy* insertions predicted by RepeatMasker, annotated protein-coding mRNA exons, and FISH probes. **(G)** RNA FISH, showing the expression of the cluster 667 in the somatic cells of a *D. mojavensis* egg chamber. Focused signals in the somatic cells are indicated by arrowheads. Scale bars = 10 μm.
(PDF)

**S7 Fig. Absence of ping-pong transposon piRNAs in *D. eugracilis* ovaries. (A–C)** Shown are the 5′ end coverage of piRNA reads (>22 nt) from *D. melanogaster* (top), *D. pseudoobscura* (middle), and *D. eugracilis* (bottom) ovaries mapping to indicated transposon sequences in counts per million genome mappers (CPM). Sense and antisense reads are coloured in black and red, respectively. Putative ping-pong pairs or absence of them are highlighted in the magnified view of the regions shown by dashed boxes (A', B', and C'). Frequencies of the 5′ overlapping bases between sense and antisense piRNAs are calculated where the characteristic 10 nt overlap is visible when there is a prominent ping-pong (A", B", and C"). The underlying data can be found in the S6 Data file.
(PDF)

**S8 Fig. *D. eugracilis* Piwi and Aubergine are bound to nearly identical pools of piRNAs. (A)** A western blotting showing the specificities of antibodies against *D. eugracilis* Piwi and Aubergine (Aub). Input ovary lysates and the elutions after the immunoprecipitation (IP) are loaded and blotted against respective antibodies. **(B–D)** Size distribution of total ovarian (B), Piwi- (C), and Aubergine-bound (D) small RNAs mapping to transposon sense (bottom) and antisense (top) sequences. The proportion of nucleotides at the 5′ end is shown by different colours, showing the Uridine preference. The abundance is normalised to the total transposon mapping reads, showing that the majority of reads are antisense for all three libraries. There are very few putative siRNAs (21 nt, marked by asterisk) compared to piRNAs (24 to 29 nt), that are only detected in the sense reads and depleted in the IP libraries. **(E and F)** Shown are the 5′ and 3′ ends of Piwi- (E) and Aubergine-bound (F) piRNAs mapping to the antisense strand of *BEL-2_Deu* from the region indicated by a dashed box in Fig 4A. The 3′ and 5′ ends of piRNAs that are one nucleotide apart, hence the putative products of phasing, are marked by arrowheads. **(E' and F')** Shown are the frequencies of Uridines found at positions relative to the 3′ ends of piRNAs mapping to *BEL-2_Deu*. +1 corresponds to the immediate downstream nucleotide position. **(E" and F")** Shown are the frequency plot of the 3′-5′ linkage of antisense *BEL-2_Deu* piRNAs. The z scores of the linkage position +1 are shown. **(G)** A box plot showing the relative abundance of genome-unique piRNAs from *D. eugracilis* mapping to 0.5 kb tiles, comparing the whole ovaries and Piwi- and Aubergine-bound pools. piRNAs mapping to the tiles from somatic clusters are enriched and depleted in Piwi- and Aubergine-bound pools, respectively. *p*-Values are calculated by Mann–Whitney *U* test. The underlying data can be

found in S6 Data and S1 Raw Images files.
(PDF)

**S9 Fig. *in-trans* ping-pong of simulated, mouse pachytene, and *Drosophila* piRNAs. (A)**
Shown at the top is the coverage of piRNA reads (>22 nt) from a mouse pachytene piRNA
cluster at chromosome 2 (PMID: 26115953). Shown at the bottom is an example of putative
phasing events. The most abundant piRNA in this region is likely made as a result of slicing by
piRNAs from distant genomic loci, which is followed by production of downstream piRNAs as
indicated by arrows. **(B)** Shown are the *in-trans* ping-pong linkage values of simulated random
piRNA pools with varying extent of ping-pong pairs artificially included (5 replicates each).
The linkage value increases by one as the proportion of artificially added ping-pong pairs
increases by 1%. **(C)** Shown are *in-trans* ping-pong linkage values of genome unique piRNAs
and all piRNA mappers from *D. pseudoobscura* and the second replicate of *D. eugracilis* ovar-
ian small RNA libraries. Estimated proportions of *in-trans* ping-pong pairs out of all piRNAs
are shown in percentage. The underlying data can be found in the S6 Data file.
(PDF)

**S1 Data. The list of genome assemblies of *Drosophila* species examined for the conserva-
tion of core piRNA pathway genes.** The first tab lists GenBank and RefSeq genome assemblies
and the second tab lists Nanopore assemblies from Miller and colleagues [57].
(XLSX)

**S2 Data. Comparison of Env protein sequences from Gypsy_DM, Gypsy12_Dpse, and Gyp-
sy_DS. (A)** Amino acid sequences of Env proteins from *Gypsy_DM*, *Gypsy12_Dpse*, and *Gyp-
sy_DS* are shown. Sequences of *Gypsy12_Dpse* and *Gypsy_DS* Env are taken from the
insertions found in the *D. pseudoobscura* and the *D.bifasciata* genomes, respectively. (**B** and **C**)
Alignments of Gypsy Env protein sequences made by Clustal Omega are shown. Functionally
important motifs are shown in boxes; from the N-terminus, signal peptide, furin cleavage site,
and the transmembrane domain [66,67]. *Gypsy_DS* Env protein was previously suspected to
lack the peptide signal and the transmembrane domain [66]. However, the genomic insertion
shown here retains those motifs.
(DOCX)

**S3 Data. The list of intact copies of *gypsy-env* retrotransposons in the *D. pseudoobscura*, *D.
bifasciata*, *D. azteca*, and *D. eugracilis* genomes.** Shown in the table are the name, chromo-
somal position, and the score of tBlastn search for each transposon insertions.
(XLSX)

**S4 Data. *aubergine* homologues found in the *obscura* group species.** The *aubergine* gene has
been duplicated or triplicated in most species of the *obscura* group. Shown are the number and
the chromosomal location of duplicated copies in each species examined.
(XLSX)

**S5 Data. Sequences of oligo DNA probes used for RNA FISH experiments.** Shown are
sequences of oligo DNA probes used for RNA FISH experiments.
(XLSX)

**S6 Data. Raw numerical data used to generate plots in the figures.** Raw numerical data for
Figs 1B, 3E, 4A–4F, 4H, S7A–S7C, S8B–S8G, S9B, and S9C are shown.
(XLSX)

**S1 Raw images. Raw images for western blots.** Original raw images for the western blots
shown in S2 and S8 Figs are shown. Regions that are used in the figures are indicated by red

dashed boxes.
(PDF)

## Acknowledgments

We thank the Brennecke lab for housing several non-*melanogaster Drosophila* strains. We thank Hiroshi Kimura and Julius Brennecke for providing us with anti RNA polymerase II antibody and anti-Armitage antibody, respectively. We thank Kazu Mochizuki and Julius Brennecke for critical comments on the manuscript.

## Author Contributions

**Conceptualization:** Rippei Hayashi.

**Data curation:** Shashank Chary, Rippei Hayashi.

**Formal analysis:** Shashank Chary, Rippei Hayashi.

**Funding acquisition:** Rippei Hayashi.

**Investigation:** Shashank Chary, Rippei Hayashi.

**Methodology:** Shashank Chary, Rippei Hayashi.

**Supervision:** Rippei Hayashi.

**Writing – original draft:** Shashank Chary, Rippei Hayashi.

**Writing – review & editing:** Shashank Chary, Rippei Hayashi.

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
