## [Editor Report · Decision Letter 0]

9 Jan 2023

Dear Dr Hayashi, 

Thank you for submitting your manuscript entitled "Mechanistic divergence of piRNA biogenesis in Drosophila" for consideration as a Research Article by PLOS Biology. Please accept my sincere apologies for the long delay in getting back to you due to the recent Christmas holiday period. 

Your manuscript has now been evaluated by the PLOS Biology editorial staff, as well as by an academic editor with relevant expertise, and I am writing to let you know that we would like to send your submission out for external peer review.

IMPORTANT: After discussions within the academic editor, we think that your manuscript would be a better fit as a Discovery Report (https://journals.plos.org/plosbiology/s/what-we-publish#loc-discovery-report), given that it remains unclear how a lack of ping-pong is resolved in D. eugracilis by an alternative pathway. Upon resubmission (details below), we would be grateful if you could tick 'Discovery Report' as the article type. 

Before we can send your manuscript to reviewers, we need you to complete your submission by providing the metadata that is required for full assessment. To this end, please login to Editorial Manager where you will find the paper in the 'Submissions Needing Revisions' folder on your homepage. Please click 'Revise Submission' from the Action Links and complete all additional questions in the submission questionnaire.

Once your full submission is complete, your paper will undergo a series of checks in preparation for peer review. After your manuscript has passed the checks it will be sent out for review. To provide the metadata for your submission, please Login to Editorial Manager (https://www.editorialmanager.com/pbiology) within two working days, i.e. by Jan 11 2023 11:59PM.

Kind regards,

Richard

Richard Hodge, PhD

Associate Editor, PLOS Biology

rhodge@plos.org

PLOS

---

## [Decision Letter · Decision Letter 1]

23 Feb 2023

Dear Dr Hayashi,

Thank you for your patience while your manuscript "Mechanistic divergence of piRNA biogenesis in Drosophila" went through peer-review at PLOS Biology. Please accept my sincere apologies for the delays that you have experienced during the peer review process. Your manuscript has now been evaluated by the PLOS Biology editors, an Academic Editor with relevant expertise, and by two independent reviewers.

In light of the reviews, which you will find at the end of this email, we are pleased to offer you the opportunity to address the comments from the reviewers in a revision that we anticipate should not take you very long. We will then assess your revised manuscript and your response to the reviewers' comments with our Academic Editor aiming to avoid further rounds of peer-review, although might need to consult with the reviewers, depending on the nature of the revisions.

In addition, I would be grateful if you could address the remaining data and other policy-related requests that I have provided below (A-F):

(A) Your manuscript is being considered as a Discovery Report, which has a maximum of 4 main figures. At this stage, we ask that you please reduce the number of main figures to 4, either by combining figure panels in the main figures or by moving some of the figures to the Supplementary. 

(B) We would like to suggest the following modification to the title, to make it more compelling for our broad readership:

“Proteins thought to be essential for piRNA biogenesis are absent in several Drosophila species that nevertheless display effective transposon restriction”

(C) You may be aware of the PLOS Data Policy, which requires that all data be made available without restriction: http://journals.plos.org/plosbiology/s/data-availability. For more information, please also see this editorial: http://dx.doi.org/10.1371/journal.pbio.1001797

- Supplementary files (e.g., excel). Please ensure that all data files are uploaded as 'Supporting Information' and are invariably referred to (in the manuscript, figure legends, and the Description field when uploading your files) using the following format verbatim: S1 Data, S2 Data, etc. Multiple panels of a single or even several figures can be included as multiple sheets in one excel file that is saved using exactly the following convention: S1_Data.xlsx (using an underscore).

-Deposition in a publicly available repository. Please also provide the accession code or a reviewer link so that we may view your data before publication.

Figure 1B, 5E, 6A-F, 6H, S7A-C, S8B-G, S9B-C

(D) Please also ensure that each of the relevant figure legends in your manuscript include information on *WHERE THE UNDERLYING DATA CAN BE FOUND*, and ensure your supplemental data file/s has a legend.

(E) We require the original, uncropped and minimally adjusted images supporting all blot and gel results reported in the following figures:

Figure S2, S8A, S11

We will require these files before a manuscript can be accepted so please prepare and upload them now. Please carefully read our guidelines for how to prepare and upload this data: https://journals.plos.org/plosbiology/s/figures#loc-blot-and-gel-reporting-requirements

(F) Please ensure that your Data Statement in the submission system accurately describes where your data can be found and is in final format, as it will be published as written there.

We expect to receive your revised manuscript within 2 months. Please email us (plosbiology@plos.org) if you have any questions or concerns, or would like to request an extension. 

**IMPORTANT - SUBMITTING YOUR REVISION**

*Resubmission Checklist*

*Published Peer Review*

*PLOS Data Policy*

*Blot and Gel Data Policy*

Kind regards,

Richard

Richard Hodge, PhD

Associate Editor, PLOS Biology

rhodge@plos.org

REVIEWS:

Reviewer #1 (Yehu Moran, signs review): The manuscript by Chary and Hayashi describes a major evolutionary change in the piRNA system of some of Drosophila species. The authors show that while some Drosophila species lost their Yb RNA helicase that is responsible for specificity of piRNA-biogenesis from the flamenco locus, an even more striking change in that system in Drosophila eugracilis is the loss of the Ago3 gene and the "ping-pong" cycle that serves for amplifying piRNAs. Altogether, these are very convincing and interesting results that demonstrate the loss an important component of a potent anti-transposon system in a lineage of flies, indicating the evolutionary plasticity of this important genomic defense system and its ability to evolve in a way that allows the latter loss of key components that might lose efficiency and probably become non-beneficial due to compensation from different anti-transposon components. In that sense I believe it can potentially fit the Discovery Report format of PLOS Biology. However, I believe it needs some revision and maybe better "packaging" to fit this format. Please find my comments below:

Major comments:

1. I believe that the evolutionary angle of this story is very much understated. For somebody who is not well familiar with the piRNA field, it might seem at a first glance like an anecdotal observation. Species lose genes all the time so what is so striking about losing Yb or Ago3? In that sense it is important to emphasize the fact that flies with no intact piRNA system tend to be sterile. Furthermore, putting these findings in the context of evolutionary "arms-race" between hosts and transposons is also important. Transposons evolve very fast, just like viruses, and hence the evolution of the systems that defend against them also tend to be very fast. In this context, losses are more frequent. Relevant papers to cite in this context include (but there are others as well):

Luo et al. 2020 BMC Evol. Biol. 20: 14; Aravin et al. 2007 Science 318: 761-764 (already mentioned by the authors but in a different context); Cosby et al. 2019 Genes Dev. 33: 1098-1116. The paper will benefit from addressing this topic in more detail in the discussion as it will make the importance of the results clearer to a broader readership.

2. The authors mention in the introduction very briefly that "Both ping-pong and phasing mechanisms are evolutionarily highly conserved, present in sponge and hydra species all the way to humans". First of all, the cited papers do not deal with "hydra species", but with cnidarians and actually Grismson et al. who the authors cite (Ref. 16) show data for Nematostella vectensis, which is a sea anemone (A very far relative of hydras. Same phylogenetic distance as between human and Drosophila). This is an important point as it helps putting the results in an appropriate evolutionary context: the ping pong cycle probably appeared in the last common ancestor of all animals. Thus, it deserves a more serious mention and might fit better in the discussion. Relevant citations that should be included in addition to Grimson et al 2008 are: Juliano et al. 2014 PNAS 111: 337-342; Praher et al. 2017 RNA Biol. 14: 1727-1741 and Calcino et al. 2018 BMC Evol. Biol. 18: 160.

3. Continuing the previous point, an important paper to mention and briefly discuss in the context of the current results is that of Lewis et al. (2018 Nat. Ecol. Evol. 2: 174-181). This paper describes the finding of somatic piRNAs in arachnids (a taxon of non-insect arthropods) and the presence and absence of the ping-pong cycle in some of the tested species. This can provide an example for another adaptation of the piRNA system over a different phylogenetic scale I the same group (arthropods).

Minor comments

1. I know that some researchers working on flies feel like Drosophila is an English word, but in fact it is in Latin. Hence, it should be italicized like any other scientific Genus name. This should be done throughout the text.

2. I feel that the last sentence of the abstract is a bit missing the truly striking result of this study: the loss of the ping-pong cycle (and Ago3) in Drosophila eugracilis and puts the focus on the loss of Yb in multiple species. Please consider to rephrase it so it will account for these findings better.

3. The authors write "We could not determine the cell-type specific expression of D. eugracilis clusters by FISH as they expressed much fewer piRNAs per kilo bases than clusters from the other species (data not shown)." I doubt that "data not shown" is a valid statement in 2023 in a journal that promotes full transparency. Please consider showing the data as supplementary.

4. Supplementary Figure 10 is in my opinion quite important in order to understand better the findings of this paper. Thus, I do not understand why it is a supplementary Figure. Please consider to make it into a main figure or combine it with an existing main figure.

Reviewer #2: Manuscript PBIOLOGY-D-22-02844R1 'Mechanistic divergence of piRNA biogenesis in Drosophila'. PIWI interacting RNAs (piRNAs) are small non-coding RNAs that target transposon transcripts for transcriptional and post-transcriptional silencing in animal gonads. The authors investigate homologs of core piRNA biogenesis factors in different closely related Drosophilids and correlate signatures of piRNA processing. They show that Yb, a co-factor of primary piRNA biogenesis in D. melanogaster, and Ago3, which is essential for secondary piRNA biogenesis during ping-pong in D. melanogaster, have been lost in some related species. Interestingly, these species show unaffected primary piRNA biogenesis and effective transposon restriction in the absence of ping-pong. The results provide an insight into conserved piRNA-guided transposon control and the variability of molecular mechanisms and co-factors. The experiments are well done and presented, and the manuscript is a pleasure to read. I strongly support publication of this work after revision.

Major points:

(1) To complete the analysis of conserved genes in different Drosophila species, the authors should include conservation analysis for zucchini, the nuclease that generates primary and phased piRNAs, and vasa, the RNA helicase that coordinates ping-pong. 

(2) Introduction: "there are two distinct modes of piR biogenesis…": to provide a complete picture for a general audience, the authors should discriminate between primary piRNA biogenesis and phasing, which is a sub-pattern of primary biogenesis or ping-pong induced ('tertiary piRNAs'). In addition to references 12 and 13, Nishimasu et al, and Ipsaro et al Nature 2012 should be cited. This would be also helpful throughout the article, as non-phased primary piRNAs in addition to phased pairs are clearly visible in Fig. 6E

(3) Fig. 1B: the authors mention a >10-fold decrease in flamenco-derived piRNAs and a > 100-fold increase in gypsy transposons in the results. The corresponding figure only shows the gypsy TE data and forgot to include the flam data. The authors should add the flam data.

(4) Fig. 4A-D: The authors focus on similarities between Armi staining in different Drosophilids in the text. However, the figures show significant differences too. For example: the ratio of germ cell compared to follicle cell staining for Armi in D. Azteca and D eugracilis. These might be differences in antibody affinities or might have biological significance. The authors should comment on this. 

(5) Chapter "Specialized piRNA biogenesis for gypsy is evolutionarily conserved in Drosophila". In addition to the co-factors for primary piRNA biogenesis in the gonadal soma (Armitage and Yb), the authors should include conservation analysis for the processing nuclease itself (Zucchini).

(6) Chapter: "Absence of ping-pong piRNAs in D. eugracilis". In support of their data revealing tolerance for loss of ping-pong in some D. eugracilis, the authors should discuss Vrettos et al., LSA 2020, who suggested that ping-pong amplification is required for germplasm and the formation of a new germline but largely dispensable for fertility in D. melanogaster. 

(7) Chapter: "Piwi and Aub receive phasing piRNAs in D. eugracilis". This should read "…primary and phased piRNAs". The authors should extend their phasing analysis to provide the following numbers: What fraction of piRNAs showed a signature of 'phased piRNA pairs' (Coincidence of a piRNA 3' end with a piRNA 5' end)? What fraction shows three or more phased piRNAs (Coincidence of a piRNA 3' end with the 5' end of a piRNA that has a 3' end that coincides with yet another 5' end). In previous reports these numbers have been rather small with fractions <1 % for ping-pong-induced phasing. The authors provide such numbers for mouse piRNAs later in the manuscript but numbers for Drosophila data are missing. Clearly defining what is meant by 'phasing' (pairs versus longer trails) and clarifying the prevalence of phased signatures in primary piRNAs and ping-pong-induced 'tertiary' piRNAs is essential to evaluate the importance of these signatures for piRNA biogenesis. 

(8) "Phasing occurs without slicing by PIWI proteins in the D. eugracilis germline". The authors should add that ping-pong independent phasing has also been observed for somatic piRNAs in D. melanogaster though at low level (Mohn et al., Science 2015; Han et al., Science 2015). Statements for mouse pachytene piRNAs should be revised in the light of new data indicating the absence of efficient slicer-induced phasing and revealing an interesting tolerance for slicer activity without complementarity base-pairing across target nt 10-11 (Dowling et al., RNA 2023). 

(9) in the discussion the authors state: "phasing biogenesis requires a 'triggering' event…" -but does it? Efficient processing by the endonuclease zucchini has been shown to produce variable and densely packed piRNAs across precursors. Individual cleavage events could produce 3' and 5' ends for neighboring piRNAs either simultaneously or from different precursor molecules. Either event would appear as 'phased' signature in piRNA sequencing data. The authors should explain the definition of 'phasing' and an observable signature in sequencing data and explain different possible mechanistic explanations. They could discuss all these possibilities in support of their observation of variable 'phasing' signatures. 

(10) The authors should include the grand-childless phenotype observed by Vrettos et al., LSA 2020 in their discussion of maternally inherited piRNAs.

Minor points:

Numbering lines and pages in the manuscript would be convenient to facilitate review.

---

## [Editor Report · Decision Letter 2]

30 Mar 2023

Dear Rippei,

On behalf of my colleagues and the Academic Editor, Rene Ketting, I am pleased to say that we can accept your manuscript for publication, provided you address any remaining formatting and reporting issues. These will be detailed in an email you should receive within 2-3 business days from our colleagues in the journal operations team; no action is required from you until then. Please note that we will not be able to formally accept your manuscript and schedule it for publication until you have completed any requested changes.

PRESS

Kind regards, 

Richard

Richard Hodge, PhD

Associate Editor, PLOS Biology

rhodge@plos.org

PLOS
